# Distinct skeletal stem cell types orchestrate long bone skeletogenesis

Thomas H Ambrosi[1], Rahul Sinha[1], Holly M Steininger[1], Malachia Y Hoover[1], Matthew P Murphy[1], Lauren S Koepke[1], Yuting Wang[1], Wan-Jin Lu[1], Maurizio Morri[2], Norma F Neff[2], Irving L Weissman[1,3], Michael T Longaker[1,4,5], Charles KF Chan[1,4]*

[1]Institute for Stem Cell Biology and Regenerative Medicine, Stanford University School of Medicine, Stanford, United States; [2]Chan Zuckerberg BioHub, San Francisco, United States; [3]Ludwig Center for Cancer Stem Cell Biology and Medicine at Stanford University, Stanford, United States; [4]Department of Surgery, Stanford University School of Medicine, Stanford, United States; [5]Hagey Laboratory for Pediatric Regenerative Medicine, Stanford University School of Medicine, Stanford University, Stanford, United States

**Abstract** Skeletal stem and progenitor cell populations are crucial for bone physiology. Characterization of these cell types remains restricted to heterogenous bulk populations with limited information on whether they are unique or overlap with previously characterized cell types. Here we show, through comprehensive functional and single-cell transcriptomic analyses, that postnatal long bones of mice contain at least two types of bone progenitors with bona fide skeletal stem cell (SSC) characteristics. An early osteochondral SSC (ocSSC) facilitates long bone growth and repair, while a second type, a perivascular SSC (pvSSC), co-emerges with long bone marrow and contributes to shape the hematopoietic stem cell niche and regenerative demand. We establish that pvSSCs, but not ocSSCs, are the origin of bone marrow adipose tissue. Lastly, we also provide insight into residual SSC heterogeneity as well as potential crosstalk between the two spatially distinct cell populations. These findings comprehensively address previously unappreciated shortcomings of SSC research.

*For correspondence:
chazchan@stanford.edu

Competing interests: The authors declare that no competing interests exist.

## Introduction

Harnessing the regenerative potential of skeletal stem cells (SSCs), as the therapeutic answer to osteoporosis, osteoarthritis, or fracture nonunions, is a long sought-after goal. Despite advances in identifying various bone marrow resident stromal cell subpopulations, translational progress has been hampered by low-resolution and inexact methodology and characterization, leading to hetero-geneous and unspecific readouts (*Bianco and Robey, 2015*; *Ambrosi et al., 2019*). To date, bone marrow stromal cells (BMSCs) with stem cell-like characteristics are mainly enriched either by plastic adherence or single fluorescence labeling in reporter mouse lines (*Kfoury and Scadden, 2015*). BMSCs have been improperly considered stem cells if they undergo osteogenic, chondro-genic, and adipogenic differentiation in vitro and express a specific set of surface markers when expanded in cultures while omitting tests for single-cell self-renewal and multipotency (*Dominici et al., 2006*). Lineage tracing mouse models have allowed spatial allocation and fate map-ping of SSC-enriched populations in genetically labeled mice either constitutively through develop-ment and adulthood, or conditionally after timepoint-specific induction (*Méndez-Ferrer et al., 2010*; *Mizoguchi et al., 2014*; *Zhou et al., 2014*; *Debnath et al., 2018*; *Mizuhashi et al., 2018*). However, each of these reporter models also marks more differentiated cell types in addition to non-skeletal lineages, thus limiting their utility as true tracers of in situ SSC activity. Recent progress

in single-cell RNA sequencing (scRNAseq) has provided a more comprehensive view of the hetero-geneity within the bone marrow microenvironment and the diversity of cell types in subpopulations of cells previously considered stem cells (*Baryawno et al., 2019*; *Tikhonova et al., 2019*; *Wolock et al., 2019*; *Baccin et al., 2020*). While those studies give a snapshot of bone marrow mes-enchymal cell diversity, they only infer in vivo functions by certain gene expression patterns known from previously described cell populations. Leptin receptor (LepR)-labeled cells, for example, have long been called SSCs, but they also label more mature cell types such as endothelial cells, pre-osteoblasts, osteoblasts, and chondrocytes of long bones (*Zhou et al., 2014*; *Tikhonova et al., 2019*). Therefore, we and others have relied on fluorescence-activated cell sorting (FACS) to pro-spectively isolate cells with unique surface marker profiles drawn from the selective expression of a broad panel of surface proteins in freshly isolated single-cell suspensions from enzymatically dissoci-ated skeletal tissue (*Sacchetti et al., 2007*; *Tormin et al., 2011*; *Chan et al., 2015*; *Ambrosi et al., 2017*; *Chan et al., 2018*). Using this approach, we have identified a mouse and human SSC with a defined lineage hierarchy of downstream progenitor cell populations and their translational value that revealed novel targets for bone and cartilage repair (*Murphy et al., 2020*; *Ambrosi et al., 2019*; *Tevlin et al., 2017*). These osteochondrogenic SSCs (ocSSCs) are restricted to bone, cartilage, and stromal lineage output but do not give rise to bone marrow adipose tissue (BMAT) (*Ambrosi and Schulz, 2017*). Strikingly, using a different set of surface markers, a perivascular SSC (pvSSC) with tri-lineage potential has been shown to be the main source of bone marrow adipocytes, implying the existence of multiple stem cell-like populations (*Ambrosi et al., 2017*). In support of this possibility, a recent scRNAseq analysis on bone marrow mesenchyme also inferred that at least two origins for osteogenic cell fates in the bone marrow exist (*Baryawno et al., 2019*).

Here, we investigated ocSSCs and pvSSCs in mice to confirm their bona fide stem cell properties and show that they are unique SSC types that are molecularly and functionally distinct.

## Results

### Postnatal long bones harbor SSC subtypes with exclusive adipogenic potential

Using FACS, we have previously identified an osteochondral SSC (CD45-Ter119-Tie2-CD51+Thy1-6C3-CD105-) that gives rise to a bone cartilage and stromal progenitor (BCSP) that in turn generates more lineage-restricted osteochondrogenic and stromal lineages (*Chan et al., 2015*). Additionally, a pvSSC (CD45-CD31-Pdgfrα+Sca1+CD24+) has been described using a different set of markers to be the source of committed adipogenic progenitor cell (APC) types and eventually all BMAT (*Figure 1A,B*; *Ambrosi et al., 2017*). To functionally compare their respective SSC properties, we freshly FACS purified equal numbers of ocSSCs and pvSSCs from newborn Actin-CreERt Rainbow mice (Rainbow mice) and transplanted each population under the renal capsule of immunodeficient NSG mice. We allowed cells to engraft and pulsed Rainbow mice with two doses of tamoxifen 3 and 4 days post-transplant. While all transplanted cells from Rainbow mice are initially green fluorescent protein (GFP)-positive, tamoxifen-induced recombination activates additional color fluorescent pro-tein expression (GFP, mCerulean, mOrange, mCherry) to enable unique genetic colorimetric labeling of single cells, which could then be assessed to track their clonal activity and lineage potential in the subsequent 2-week interval. We observed high clonal activity in transplants with ocSSCs and pvSSCs, which is a hallmark of stem cell activity (*Figure 1C*), as well as clonally derived osteochondrogenic and osteochondroadipogenic cell types in ocSSC and pvSSC grafts, respectively (*Figure 1—figure supplement 1A*). We next transplanted freshly FACS-purified GFP-labeled ocSSCs and pvSSCs from newborn mice under the renal capsule of NSG mice to assess heterotopic bone formation in vivo. 4-week grafts of both cell populations formed ossicles with bone and cartilage capable of generating an ectopic hematopoietic niche through recruitment of host-derived blood and immune cells (*Figure 1D,E* and *Figure 1—figure supplement 1B*). As expected, only pvSSC-derived grafts con-tained adipocytes. To confirm the long-term self-renewal of SSCs, we dissociated 4-week grafts for FACS analysis and found, in line with in vitro expanded cells, that phenotypic ocSSCs and pvSSCs remained present (*Figure 1—figure supplement 1C*). Fibroblast colony-forming unit (CFU-F) and tri-lineage differentiation assays on freshly bulk-isolated ocSSCs and pvSSCs also showed high clono-genic and osteochondrogenic differentiation potential in vitro (*Figure 1—figure supplement 1D,E*).

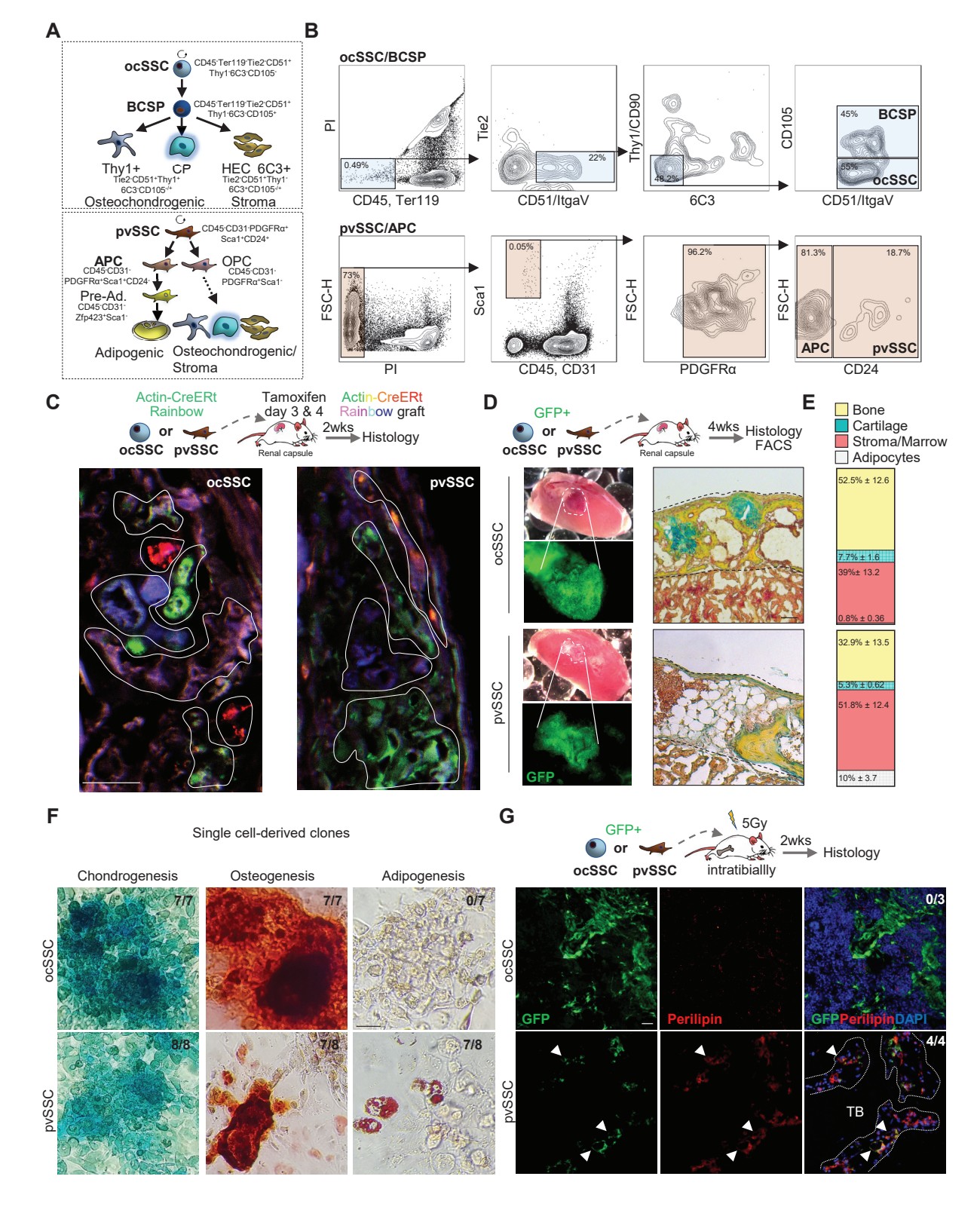

**Figure 1.** Two cell populations with skeletal stem cell characteristics in postnatal long bones. (**A**) Diagram showing two previously described skeletal stem cell (SSC) populations and the downstream populations they generate, which were defined by the specific expression patterns of cell surface proteins. Top: SSC lineage tree of the osteochondral SSC (ocSSC) that gives rise to bone, cartilage, and stromal populations. Bottom: the lineage tree of the perivascular SSC (pvSSC) able to give rise to bone, cartilage, adipose tissue, and stromal populations. BCSP: bone cartilage stroma progenitor;

*Figure 1 continued on next page*

*Figure 1 continued*

CP: cartilage progenitor; APC: adipogenic progenitor cell; Pre-Ad.: pre-adipocyte; OPC: osteochondrogenic progenitor cell. (B) Representative flow cytometric gating strategy for the isolation of ocSSCs (top) and pvSSCs (bottom). (C) Representative confocal microscopy images of in vivo derived single-color clonal colonies of renal capsule-transplanted purified ocSSCs (left) and pvSSCs (right) derived from Actin-CreERt Rainbow mice. Three independent transplants per cell type under renal capsules were performed. (D) Renal capsule transplant-derived ossicles of purified GFP-labeled ocSSCs (top) and pvSSCs (bottom). Images show the photograph of the kidney with transplant (top left) and GFP signal of graft tissue (bottom left) as well as Movat pentachrome cross-section staining (right). (E) Quantification of ocSSC (top) and pvSSC (bottom) graft composition. Results of three separate experiments with n = 3 per SSC type. All data are shown as mean ± SEM. (F) Representative images of staining of clonally derived cultures that underwent tri-lineage differentiation assays in vitro. Alcian blue (chondrogenesis), Alizarin red S (osteogenesis), and oil red O (adipogenesis) stainings are shown along with the number of clones that stained positive for each differentiation type (ocSSC n = 7; pvSSC n = 8 clones). (G) Representative immunohistochemistry images for GFP (green) and perilipin (red) of tissue derived from intratibially transplanted purified GFP-labeled ocSSCs (top) and pvSSCs (bottom) 2 weeks after injection. White arrowheads: GFP$^+$Perilipin$^+$ cells. TB: trabecular bone. Three separate experiments with ocSSC n = 3 and pvSSC n = 4. Scale bars, 30 μm.

The online version of this article includes the following source data and figure supplement(s) for figure 1:

**Figure supplement 1.** Perivascular SSCs, but not ocSSCs, are a source of bone marrow adipose tissue.

**Figure supplement 1—source data 1.** Functional characterization of ocSSCs and pvSSCs.

Strikingly, in comparison to pvSSCs, ocSSCs did not acquire adipogenic potential even under strong adipogenic stimuli and when isolated from BMAT-rich long bones of 2-year-old mice (*Figure 1—figure supplement 1F*). This difference in lineage potential was also observed in single-clone CFUs derived from individual sorted ocSSCs and pvSSCs (*Figure 1F*). Finally, we traced the lineage output of ocSSCs and pvSSCs in their endogenous environments in vivo. We sublethally irradiated NSG mice to enhance engraftment and injected freshly sorted GFP-labeled SSC populations into the intramedullary space of the tibia. 2 weeks later, tibia bones were sectioned and immunohistologically stained for osteogenic (osteocalcin), chondrogenic (collagen 2), and adipogenic (perilipin) fates. Again, osteochondrogenic lineage output was observed in both cell types, and only pvSSCs generated perilipin-positive bone marrow adipocytes (*Figure 1G* and *Figure 1—figure supplement 1G–I*). Taken together, ocSSCs and pvSSCs display bona fide stem cell characteristics including high clonogenicity, long-term self-renewal, and multi-differentiation capacity in vitro and in vivo but show disparate adipogenic potential.

## Osteochondral and perivascular SSCs are distinct skeletal stem cell populations

SSCs have been shown to occupy specific microenvironments. In agreement with the original reports, we found that ocSSCs/BCSPs are enriched in micro-dissected non-marrow fractions of femurs while pvSSCs and the committed APCs are more evenly distributed but accumulate at the ends of long bones where coincidentally BMAT first appears (*Figure 2A* and *Figure 2—figure supplement 1A*; *Chan et al., 2015*; *Scheller et al., 2015*; *Ambrosi et al., 2017*). Of note, both SSC types were found on the periosteum, a site described to harbor a distinct stem cell for bone regeneration (*Debnath et al., 2018*; *Duchamp de Lageneste et al., 2018*). Flow cytometric analysis of long bones at different timepoints during skeletal maturation further revealed that ocSSCs are present in high abundance at the early stages of limb development (E13.5), linking them to bone formation. Contrastingly, pvSSCs are first detectable around E15.5, similar to hematopoietic stem cell (HSC) detection in mouse bone marrow, and peak perinatally (*Figure 2B*; *Rowe et al., 2016*). To explore the temporal connection between pvSSC and adipocyte emergence, we tested whether the absence of pvSSCs precludes appearance of BMAT formation. To that end, we isolated long bones from ubiquitously GFP-expressing mice at E12-E13.5, during which they are enriched with ocSSCs, as well as from postnatal day (P) 1–10 at which stage pvSSCs become more prominent. We then transplanted dissected bones from these different timepoints under the renal capsule of young NSG mice for 4 weeks. Interestingly, transplanted ≤E13.5 tibias failed to develop GFP-labeled diaphyseal bone marrow adipocytes whereas ≥P1 tibia bones strongly accumulated adipocytes below the growth plate in this experimental setting (*Figure 2C* and *Figure 2—figure supplement 1B–D*), corresponding to the difference in the frequency of ocSSCs vs pvSSCs between these stages. To further rule out that ocSSCs and pvSSCs are overlapping cell populations, we included Sca1, one of the positive markers used to purify pvSSCs, to our ocSSCs/BCSPs surface marker profile

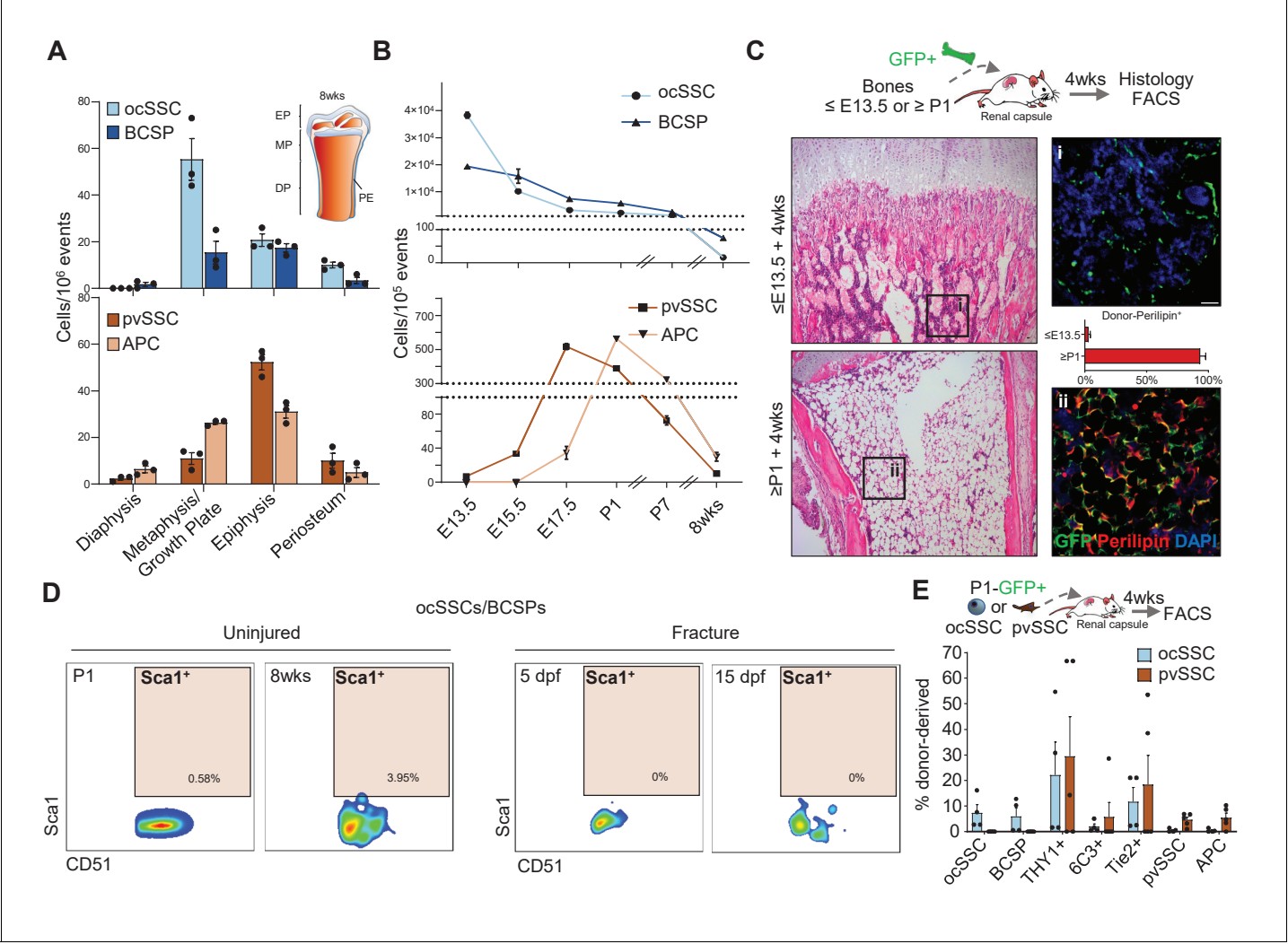

**Figure 2.** Osteochondral and perivascular SSCs are anatomically and developmentally distinct. (**A**) Flow cytometric quantification of micro-dissected long bone regions of 8-week-old mice for the prevalence of osteochondrogenic skeletal stem cells (ocSSCs), bone cartilage and stromal progenitors (BCSPs), perivascular SSCs (pvSSCs), and adipogenic progenitor cells (APCs). EP: epiphysis, MP: metaphysis, DP: diaphysis, PE: periosteum (n = 3 mice). (**B**) Frequency of ocSSCs, BCSPs (top) and pvSSCs, APCs (bottom) in the long bones of mouse embryos developing into postnatal life assessed by flow cytometry (n = 3 mice per age group). (**C**) Dissected limb bones of GFP-expressing embryos between E12-E13.5 (≤E13.5) and postnatal day 1–10 (≥P1) were transplanted under the renal capsule of NSG mice. 4 weeks after transplantation, bones were dissected out and sectioned for hematoxylin and eosin staining (left) and immunohistological staining for GFP (green) and perilipin (red) quantified as the percentage of GFP⁺Perilipin⁺ (right) cells (n = 4 bones per age group) below the distal growth plate. (**D**) Representative flow cytometric plots showing Sca1 expression in cells gated for CD45⁻Ter119⁻Tie2⁻CD51⁺6C3⁻Thy1⁻ (ocSSC/BCSP) in long bones at postnatal day 1 and at 8 weeks of age (uninjured) as well as during regeneration at days post fracture (dpf) 5 and 15. (**E**) Flow cytometric analysis of lineage output in ocSSC (n = 4)- and pvSSC (n = 5)-derived renal grafts displayed as the percentage of all donor-derived cells from at least two independent experiments. All data are shown as mean + SEM. Scale bars, 30 μm.

The online version of this article includes the following source data and figure supplement(s) for figure 2:

**Source data 1.** Anatomical and developmental assessment of ocSSCs and pvSSCs.

**Figure supplement 1.** No overlap between bone-resident SSC subtypes.

**Figure supplement 1—source data 1.** Non-overlapping SSC subtypes.

for flow cytometric analysis. We found that Sca1 expression was virtually absent in newborn and 8-week-old uninjured long bone as well as post-fracture day-5 and -15 ocSSCs/BCSPs (*Figure 2D*). Analysis of the ocSSC lineage tree revealed that the majority of Sca1-positive cells were contained in the Tie2-positive fraction, in agreement with our earlier work showing adipogenic potential of that population (*Figure 2—figure supplement 1E,F*; *Chan et al., 2013*). Importantly, ocSSCs from E13.5 and newborn mice did not give rise to pvSSCs/APCs in 4-week renal grafts, and pvSSC-derived

grafts did not generate ocSSCs/BCSPs (*Figure 2E* and *Figure 2—figure supplement 1G*). Altogether, these results indicate that two distinct SSC types with specific developmental occurrence and anatomical distribution exist in postnatal mouse long bones.

## SSC diversity serves distinct niche functions

Having established the existence of the two SSC populations, we next asked how they were molecularly and functionally diverse. We conducted SmartSeq2 scRNAseq analysis on a dataset filtered for high-quality ocSSCs (143 cells) and pvSSCs (169 cells) from young adult mice (*Figure 3—figure supplement 1A–C*; *Picelli et al., 2014*). Strikingly, ocSSC and pvSSC clustered separately with *Ly6a/ Sca1* expression pattern, confirming cell-type specificity (*Figure 3A* and *Figure 3—figure supplement 1D*). Gene ontology (GO) analysis of differentially expressed genes between both cell types through GO Biological Processes demonstrated a strong association of ocSSCs with skeletal development and formation, while pvSSC genes were linked to extracellular matrix organization and regulation of hematopoietic cell types (*Figure 3B,C*). The ocSSC population also expressed much higher levels of commonly known osteochondrogenic genes (*Acan*, *Col2a1*, *Pthr1*, *Spp1*), whereas pvSSCs exhibited elevated gene counts for fibroblast markers (*Pdgfra*, *Postn*, *Dpt*) and factors related to hematopoietic interactions (*Cxcl12*, *Igf1*, *Rarres2*) (*Figure 3D*). Collectively, these findings suggested that ocSSCs are a major source of bone-forming cells while pvSSCs might contribute to specific niche environments in long bones. To test this functionally, we sublethally irradiated mice to compromise hematopoietic niches and assessed cell frequencies in long bones via flow cytometry 2 weeks later. pvSSCs and their downstream APCs were strongly increased compared to non-irradiated mice in line with their potential role in supporting hematopoietic recovery and the increase in BMAT seen upon radiation (*Naveiras et al., 2009*), while irradiation-sensitive ocSSC/BCSPs showed a decline in numbers (*Figure 3E* and *Figure 3—figure supplement 1E*). In contrast, when we assessed the prevalence of SSC types during bi-cortical fracture regeneration, we found ocSSCs to accumulate significantly more in callus tissue compared to pvSSCs as expected from their pivotal role in skeletal repair (*Figure 3F* and *Figure 3—figure supplement 1F*; *Marecic et al., 2015*). Since aging is known to drive bone loss and BMAT accumulation in mouse bones, we compared pvSSC and ocSSC cell frequencies in 2- and 30-month-old mice. While bone anabolic ocSSCs were significantly reduced with age, bone marrow adipocytes forming pvSSCs/APCs were increased (*Figure 3G*). Finally, co-transplantation of equal numbers of purified, uniquely labeled SSC types under the renal capsule of NSG mice generated ossicles with ocSSCs as a key contributor to osteochondrogenic tissue and reticular cells, whereas pvSSC-derived cells were more prevalent in areas anatomically close to host-derived hematopoietic tissue (*Figure 3H*). In sum, these findings suggest different functional roles between osteochondral and pvSSCs and that changes in their abundance correspond to disturbances in skeletal homeostasis.

## Commonly used single gene cell labels do not faithfully mark pure skeletal stem cells

Reporter mouse models are a crucial tool for the study of bones, including stem cell-based skeletal processes. However, available lineage tracers merely enrich for SSC populations (*Ambrosi et al., 2019*; *Tikhonova et al., 2019*) and, therefore, conclusions drawn need to be carefully put into perspective (*Figure 4A*). scRNAseq data suggested that genes reported to describe SSC populations, for example, *Pthrp*, *Ctsk*, *Cxcl12*, and *Osx*, show variable expression within and between ocSSCs and pvSSCs (*Figure 4B*; *Greenbaum et al., 2013*; *Mizoguchi et al., 2014*; *Mizuhashi et al., 2018*; *Debnath et al., 2018*). Similarly, reporter strains for labeling more mature cell types such as chondrocytes (*Col2a1*, *Acan*) and osteoblasts (*Col1a1*, *Spp1*) also show expression in stem cell-like populations (*Figure 4B*). Recently, LepR expression has been shown to strongly enrich for bone CFU-F although bulk-sequencing and scRNAseq strongly suggest that *LepR*-expressing populations are highly heterogeneous (*Zhou et al., 2014*; *Tikhonova et al., 2019*; *Figure 4A,B*). When we analyzed LepR expression on ocSSC/pvSSC lineage populations by FACS, we found LepR expression in both cell types (*Figure 4C*). Separating ocSSCs or pvSSCs into LepR-positive and -negative fractions did not alter CFU-F capacity (*Figure 4D*) and potential to give rise to osteocalcin-expressing osteoblasts in vitro (*Figure 4E–F*). These results highlight the greater ability of flow cytometry to resolve

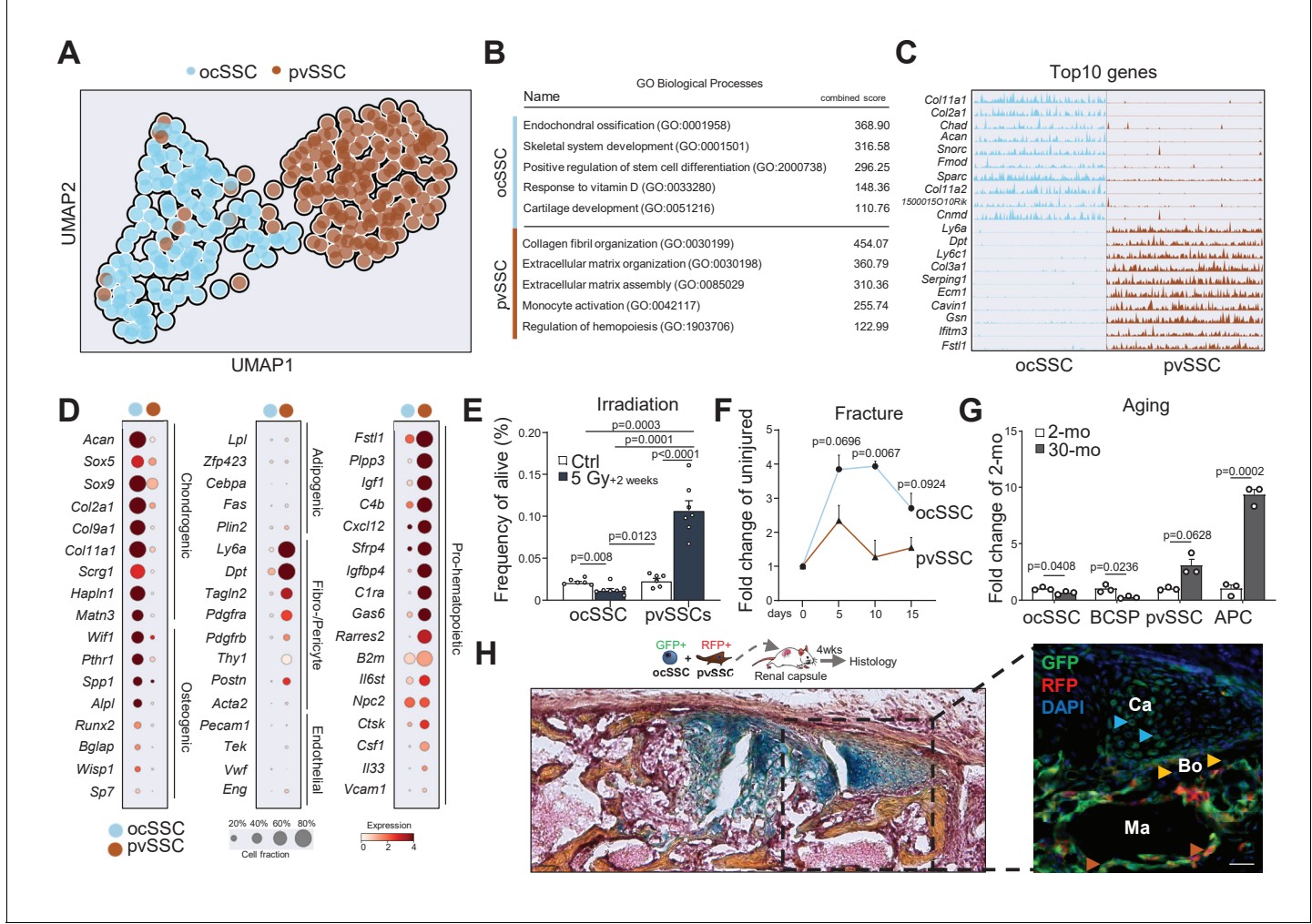

**Figure 3.** Molecular differences of SSC types infer specific niche functions. (**A**) Single-cell RNA-sequencing analysis results of 143 osteochondrogenic skeletal stem cells (ocSSCs) and 169 perivascular SSCs (pvSSCs) shown as clustering by Uniform Manifold Approximation and Projection (UMAP). (**B**) Top Gene Ontology (GO) Biological Processes by all differentially expressed genes between ocSSCs and pvSSCs as determined through EnrichR. (**C**) Track plots of top 10 differentially expressed genes in ocSSCs and pvSSCs showing individual peaks per cell as the degree of their expression. (**D**) Dot plots showing expression of selected genes in ocSSCs and pvSSCs previously reported to characterize specific cell types. (**E**) Flow cytometric quantification of ocSSCs and pvSSCs 2 weeks after whole-body sublethal irradiation of 8-week-old mice (n = 6 control; n = 7 5Gy, from two independent experiments). (**F**) Flow cytometric quantification of ocSSCs and pvSSCs at various days after stabilized bi-cortical femoral fractures of 8-week-old mice shown as fold change of uninjured (n = 3 per timepoint from two independent experiments). (**G**) Flow cytometric quantification of ocSSCs, bone cartilage and stromal progenitors (BCSPs), pvSSCs, and adipogenic progenitor cells (APCs) in long bones of young (8 weeks; n = 3) and aged (30 months; n = 3) mice shown as fold change of young. (**H**) Renal capsule-derived grafts of co-transplants of equal numbers of ocSSCs and pvSSCs showing Movat pentachrome-stained cross-section (left) and the corresponding immunohistological staining (right) for GFP (ocSSC-derived cells) and red fluorescent protein (RFP) (pvSSC-derived cells). Light blue arrowhead: cartilage (Ca); yellow arrowhead: bone (Bo); brown arrowhead: marrow (Ma) lining cells. All data are shown as mean + SEM. Significance between groups was assessed by unpaired, two-tailed Student's t-test and corrected with Welch's test for unequal distribution if needed. Scale bars, 30 μm.

The online version of this article includes the following source data and figure supplement(s) for figure 3:

**Source data 1.** Changes in SSC abundance in response to injury and aging.

**Figure supplement 1.** Two genetically and functionally distinct SSC lineages of long bones.

**Figure supplement 1—source data 1.** Quality control of single cell RNA-sequencing data.

functionally distinct SSC variants by simultaneously tracking expression of a panel of surface proteins over reporter mouse models that rely on the selective expression of a single gene.

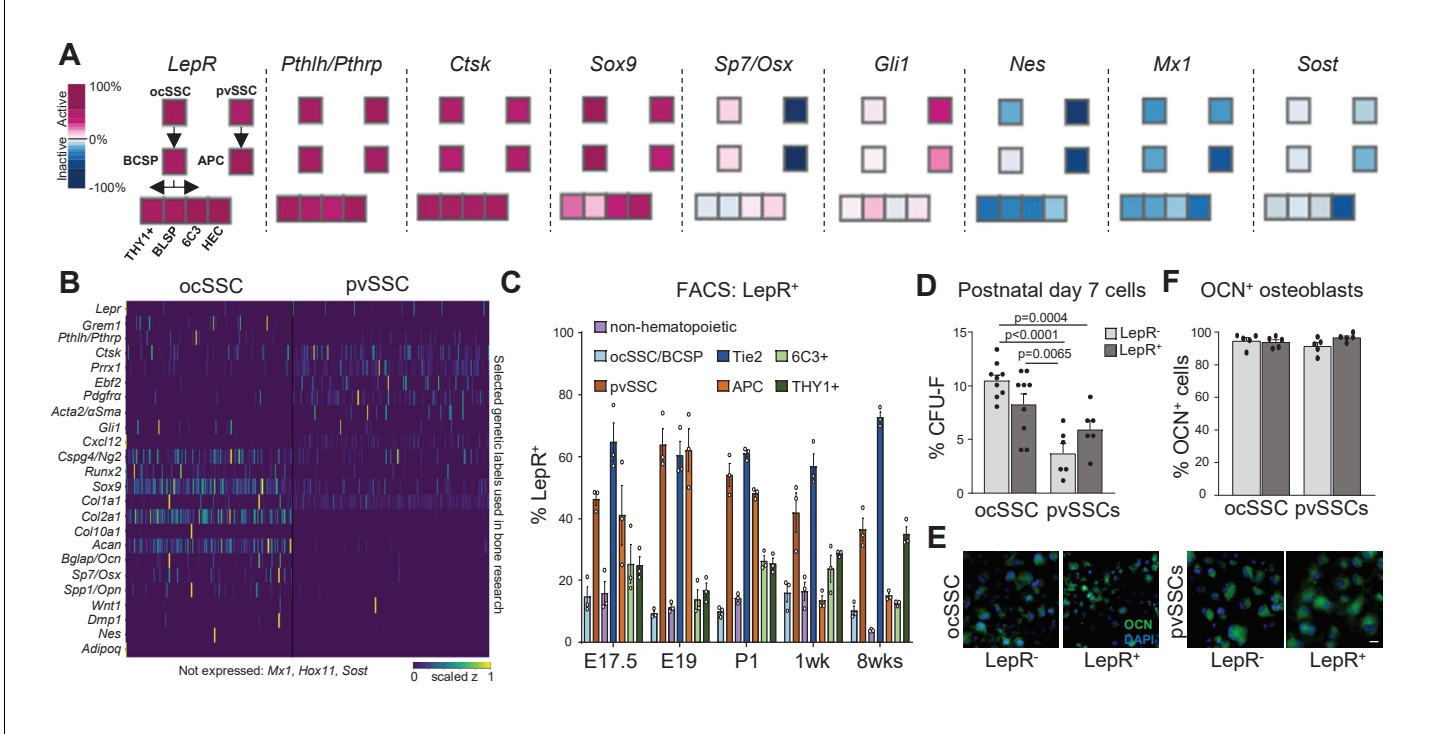

**Figure 4.** Variable expression of commonly used reporter genes in skeletal stem cell types. (**A**) Microarray data of freshly purified bulk cell populations of the previously defined osteochondrogenic skeletal stem cell (ocSSC) and perivascular SSC (pvSSC) lineage trees from 8-week-old mice showing gene expression of markers commonly known to trace and/or label cell populations enriched for SSCs. Expression is shown as normalized activity as processed by GEXC (Gene Expression Commons). Each cell-type represents a pool of cells derived from three to four mice. (**B**) Single-cell heatmaps of gene expression of markers commonly used to trace and/or label cell populations enriched for SSCs as observed in single-cell RNA-sequencing (scRNAseq) results of ocSSCs and pvSSCs. (**C**) Flow cytometric analysis of antibody labeling for leptin receptor (LepR) in different populations of the ocSSC and pvSSC lineage trees in differently aged mice (n = 3). (**D**) Fibroblast colony-forming unit (CFU-F) assay of freshly isolated ocSSCs and pvSSCs separated by their expression of LepR (ocSSC n = 9; pvSSC n = 6, from two independent experiments). (**E**) Expression of osteocalcin (OCN, green) in in vitro osteogenically differentiated ocSSCs and pvSSCs in LepR-positive and -negative fractions. (**F**) Quantification of percentage of cells expressing OCN as determined by antibody labeling at 2 weeks of osteogenic differentiation (n = 5). Significance between groups was assessed by unpaired, two-tailed Student's t-test. All data are shown as mean + SEM. Scale bars, 30 µm.

The online version of this article includes the following source data for figure 4:

**Source data 1.** Differences of LepR expression in SSC subtypes.

## Stem cell crosstalk by distinct SSC subpopulations facilitates niche interactions

To assess the diversity within ocSSC and pvSSC variants, we conducted Leiden clustering on our scRNAseq dataset. This yielded three subclusters for ocSSCs while pvSSCs were homogeneous in their overall gene signature (*Figure 5A*). Looking at the top differentially expressed genes between the four clusters, we found that clusters 3 and 4 were enriched for marker genes of chondrogenic and osteogenic fates, respectively (*Figure 5B,C*). Further, cluster 1 (pvSSCs) and cluster 2 were enriched for stem cell-associated genes (*Figure 5C* and *Figure 5—figure supplement 1A*), whereas clusters 3 and 4 showed pathway enrichment related to active bone formation processes (*Figure 5—figure supplement 1B*). Cell cycle status and CytoTrace analysis additionally inferred that osteo-chondrogenic SSC-enriched clusters were more active (*Figure 5—figure supplement 1C,D*), alto-gether suggesting that ocSSCs contained a larger fraction of cells pre-primed to commit to specific fates, potentially reflecting their primary role in constantly maintaining and remodeling bones. As we observed that both SSC types can form ectopic bone, complete with hematopoietic marrow, and that co-transplantation seems to result in a defined, coordinated lineage output (*Figures 1D* and *3H*), we next looked for possible signaling interactions between ocSSCs and pvSSCs. Leiden clusters revealed patterns in expression of combinations of ligands and cognate receptors that are highly

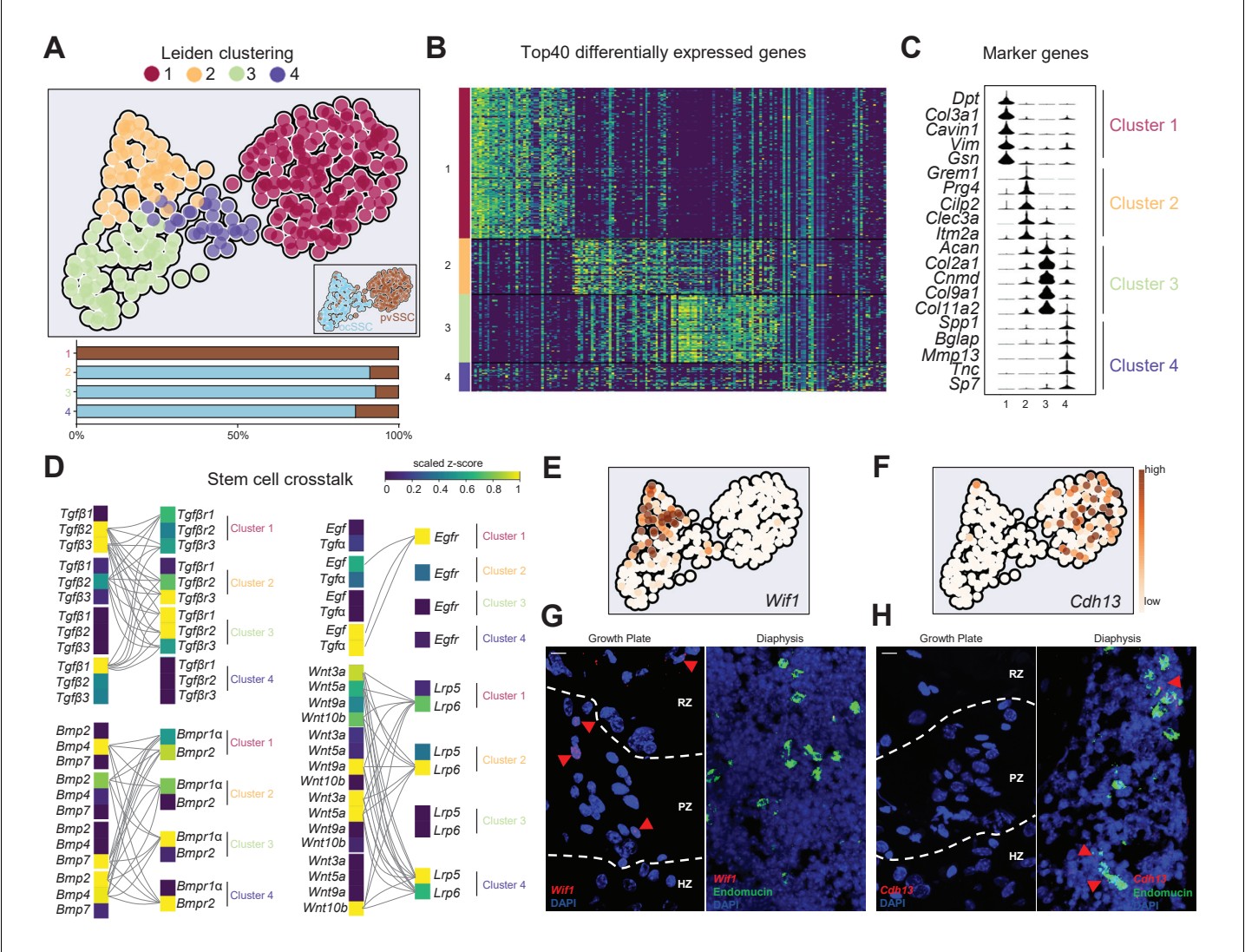

**Figure 5.** Stem cell crosstalk by distinct SSC subpopulations facilitates niche interactions. (A) Single-cell RNA-sequencing analysis results of osteochondrogenic skeletal stem cells (ocSSCs) and perivascular SSCs (pvSSCs) shown as Leiden clustering to reveal heterogeneity within cell populations (top). Composition of clusters by SSC type (bottom). Cluster 1: 155 cells; cluster 2: 57 cells; cluster 3: 70 cells; cluster 4: 30 cells. (B) Heatmap of top 40 differentially expressed genes for each of the four Leiden clusters. (C) Violin plots of selected marker genes for each of the four Leiden clusters. (D) Expression of selected ligands and their receptors in Leiden clusters of SSC populations. Connected ligand-receptor genes in pairs have scaled z-score expression >0.5. (E) Expression of cluster 2-specific marker *Wif1* in UMAP plot. (F) Expression of cluster 1-specific marker *Cdh13* in UMAP plot. (G) Representative in situ RNAscope images showing detection of the *Wif1* RNA transcripts in the growth plate (left) and their absence in diaphyseal bone marrow (right). (H) Representative in situ RNAscope images showing detection of the *Cdh13* RNA transcripts in diaphyseal bone marrow (right) and their absence in the growth plate (left). Red arrowheads: RNA transcript-expressing cells. RZ: resting zone; PZ: proliferative zone; HZ: hypertrophic zone. Scale bars, 10 μm.

The online version of this article includes the following source data and figure supplement(s) for figure 5:

**Figure supplement 1.** Single-cell heterogeneity and crosstalk of SSC subtypes.

**Figure supplement 1—source data 1.** Cytotrace analysis of leiden clusters.

suggestive of crosstalk between ocSSCs and pvSSCs (*Figure 5D* and *Figure 5—figure supplement 1E*). For example, cells of cluster 1 (pvSSCs) expressed *Tgfb2/Tgfb3* and *Wnt* ligand genes, well-known pro-chondrogenic and -osteogenic signals, which might support the specific fate commitment of ocSSCs enriched in clusters 2–4 that expressed respective receptors (*Figure 5D*). Reversely, *Bmp*, *Tgfa*, and *Egf* ligands from cells of primed clusters 3 and 4 might also signal back to pvSSCs, which express the canonical cognate receptor genes to these pathways, implying feedback regulation

across SSC types (*Figure 5D*). Finally, previous studies have mapped ocSSCs to the growth plate and pvSSCs to perivascular bone marrow regions using in situ localization (*Chan et al., 2015*; *Ambrosi et al., 2017*). Taking advantage of our scRNAseq dataset, we identified *cadherin-13* (*Cdh13*), the receptor for the adipogenic factor adiponectin (*Hug et al., 2004*), and *Wnt inhibitory factor-1* (*Wif1*), previously identified to be expressed in a small subset of BMSCs (*Tikhonova et al., 2019*), as new marker genes for the most stem cell-like clusters 1 (pvSSC) and 2 (ocSSC), respectively (*Figure 5E,F*). In situ hybridization by RNAscope confirmed a restricted expression of *Wif1* in the resting and proliferative zones of the growth plate, while *Cdh13* was absent and found in cells located in the bone marrow close to endomucin-expressing vasculature (*Figure 5G,H*). As observed via flow cytometric analysis, *Wif1*- and *Cdh13*-expressing cells were also found among periosteal cells (*Figure 5—figure supplement 1F,G*). In summary, these results confirm the distinct and overlapping anatomical localization of SSC variants in long bones and provide a glimpse into the complex crosstalk between them.

## Discussion

The current knowledge of stem cell-dependent processes for maintaining skeletal function and regeneration is based on imprecise approaches. Using a rigorous procedure to establish SSC identities and properties, our presented work here provides a new perspective into the complexity of bone tissue and how its integrity is facilitated at the stem cell level. The description of two SSC types with distinct differentiation potentials challenges the general assumption of bifurcation choices for osteogenesis versus adipogenesis of so-called 'mesenchymal stem cells' (MSCs) (*Figure 5—figure supplement 1H*).

The osteochondral SSC, closely associated with marrow-free spaces such as the growth plate or periosteum, was highly prevalent in early limb development, remained present in low numbers throughout adulthood, and could be activated to accumulate upon injury. We also showed that ocSSCs never gave rise to BMAT and may be the main source of osteochondrogenic tissue in long bones. Therefore, ocSSCs might be the SSC type exclusively targeted by the *Osx-Cre* reporter line that has been previously described to label cells that form bone and transient stromal cells in the fetal skeleton, while being restricted to osteolineages during adulthood (*Mizoguchi et al., 2014*). Follow-up studies will have to more closely examine the molecular as well as functional overlap and differences of ocSSCs inhabiting various long bone niches, such as the growth plate, periosteum, and articular surface (*Duchamp de Lageneste et al., 2018*; *Newton et al., 2019*; *Murphy et al., 2020*). Based on recent reports, ocSSCs are included as subpopulations of Pthrp-expressing growth plate and Ctsk-expressing periosteal stem and progenitor cell types (*Debnath et al., 2018*; *Mizuhashi et al., 2018*). Our in situ mapping of ocSSCs confirms the presence within the resting and proliferative growth plate zones where they might give rise to downstream lineages through hypertrophic chondrocyte intermediates. More sophisticated single-cell in vivo tracing will be needed to assess to what extent ocSSCs include or generate these hypertrophic chondrocytes that have been suggested to be a source of osteogenic and stromal marrow cell types (*Yang et al., 2014a*; *Yang et al., 2014b*; *Tan et al., 2020*).

pvSSCs, demonstrated to be essential for BMAT development, occurred at a later embryonic stage in sync with the homing of HSCs to bone marrow niches (*Figure 2B–C*; *Rowe et al., 2016*). A critical role for mesenchymal cell types to maintain HSC quiescence has long been established (*Ding et al., 2012*). When intratibially injected, pvSSCs also seemed to distribute more evenly throughout the marrow, potentially to HSC niches, whereas ocSSC-derived cells mainly aggregated in a more confined space (*Figure 1—figure supplement 1G–I*). In line with our scRNAseq expression data on pvSSCs, recent scRNAseq studies have found abundant adipo-primed cell populations in the bone marrow that were implied to be a major reservoir of pro-hematopoietic, pro-vascular, and anti-osteogenic factors (*Zhong et al., 2020*; *Tikhonova et al., 2019*). Finally, in contrast to ocSSCs, pvSSCs and their downstream APCs were increased in frequency in older mice (*Figure 3G*) and thus correlate with the observation of higher HSC abundance during aging (*Pang et al., 2011*). Future work will have to provide functional evidence for the interconnection of the pvSSC and HSC lineages as well as the potential roles of pvSSCs/APCs in age-related myeloid skewing of HSCs.

Previous work has established a non-hematopoietic, non-endothelial, Prrx-1 mesenchymal origin of pvSSCs (*Ambrosi et al., 2017*). It remains unclear, however, whether pvSSCs arise as a

subpopulation of a bone-resident cell population or whether they immigrate through the circulation or from surrounding tissues. One possibility is that pvSSCs originated from osteo-plastic perivascular mural cell and pericyte populations that had ingressed into the bone marrow space during endo-chondral ossification in limb formation (*Pearson et al., 1986*; *Crisan et al., 2008*; *James et al., 2010*; *Corselli et al., 2012*). Although these perivascular lineages may not possess inherent skeleto-genic activity prior to their invasion of bone tissues, their subsequent development in the bone mar-row space in close association with cells that express skeletogenic morphogens such as Bone Morphogenetic Protein (BMP) and hedgehog may have primed them toward skeletal fates (*Urist, 1970*; *Regard et al., 2013*; *Chan et al., 2015*; *Salazar et al., 2016*). Although we cannot exclude the plasticity between ocSSCs and pvSSCs, our experiments conducted here do not suggest the interconversion between the two cell types. Yet, specific stimuli such as high levels of Bmp2 or Wnt might be able to convert pvSSCs into ocSSC-like cells (*Chan et al., 2015*; *Matsushita et al., 2020*). Available mouse strains, for example the LepR-Cre reporter line, are not able to answer these open questions as we observed that the expression of tracer genes did not functionally separate SSC types. One obvious reason is that loxP-site excision-directed fluorescence reporters perma-nently stay on, not necessarily reflecting the current expression of the gene driving Cre-recombinase. Marker expression is dependent on the developmental stage of a given cell and is not restricted to the stem cell state as it also labels derivative downstream cell populations. In contrast, targeting SSCs by prospectively isolating them with a combination of surface markers gives a more faithful snapshot of a given cell population, however, at the expense of not being able to fate map rare cells in situ or directly test key molecular functions. While our approach provides a significant advance toward highly homogeneous SSC populations, scRNAseq analysis also indicated a remaining degree of cellular heterogeneity that will have to be further functionally resolved in future studies (*Figure 5A*). Nonetheless, the new results on SSC subtypes that we now present offer a vantage point to identify more explicit markers for novel SSC reporter mouse models that will mitigate cur-rent shortcomings.

In conclusion, these findings comprehensively describe the existence of two distinct SSC popula-tions in postnatal long bones of mice that might regulate each other through cellular crosstalk. Excit-ingly, these findings may be readily translatable to the human setting through existing reports of SSC populations that have comparable characteristics (*Sacchetti et al., 2007*; *Chan et al., 2018*).

# Materials and methods

**Key resources table**

| Reagent type (species) or resource | Designation | Source or reference | Identifiers | Additional information |
|---|---|---|---|---|
| Strain | Mouse: B6 (C57BL/Ka-Thy1.1-CD45.1) | The Jackson Laboratory | JAX: 000406 | RRID:IMSR_JAX:000406 |
| Strain | Mouse: NSG (NOD.Cg-Prkdcscid Il2rgtm1Wjl/SzJ) | The Jackson Laboratory | JAX: 005557 | RRID:IMSR_JAX:005557 |
| Strain | Mouse: GFP (C57BL/6-Tg(CAG-EGFP)1Osb/J) | The Jackson Laboratory | JAX: 003291 | RRID:IMSR_JAX:003291 |
| Strain | Mouse: Actin-CreERt Rosa26-Rainbow (homozygous) | In-house | N/A | |
| Antibody | Anti-Mouse Ly-6A/E (Sca-1) APC (clone: D7) | ThermoFisher | Cat#: 17–5981 | FACS 1:200 (RRID:AB_469487) |
| Antibody | Anti-Mouse Ly-6A/E (Sca-1) Alexa Fluor 700 (clone: D7) | ThermoFisher | Cat#: 56–5981 | FACS 1:200 (RRID:AB_657837) |
| Antibody | Anti-Mouse Ly-6A/E (Sca-1) PE/Cy7 (clone: D7) | ThermoFisher | Cat#: 25–5981 | FACS 1:200 (RRID:AB_469669) |
| Antibody | Anti-Mouse CD45 FITC (clone: 30-F11) | ThermoFisher | Cat#: 11–0451 | FACS 1:200 (RRID:AB_465050) |

*Continued on next page*

*Continued*

| Reagent type (species) or resource | Designation | Source or reference | Identifiers | Additional information |
|---|---|---|---|---|
| Antibody | Anti-Mouse CD45 APC (clone: 30-F11) | ThermoFisher | Cat#: 17–0451 | FACS 1:200 (RRID:AB_469393) |
| Antibody | Anti-Mouse CD45 PE/Cy5 (clone: 30-F11) | ThermoFisher | Cat#: 15–0451 | FACS 1:200 (RRID:AB_468751) |
| Antibody | Anti-Mouse CD31 (PECAM-1) FITC (clone: 390) | ThermoFisher | Cat#: 11–0311 | FACS 1:200 (RRID:AB_465011) |
| Antibody | Anti-Mouse CD31 (PECAM-1) APC (clone: 390) | ThermoFisher | Cat#: 17–0311 | FACS 1:200 (RRID:AB_657735) |
| Antibody | Anti-Mouse TER-119 PE/Cy5 (clone: TER-119) | ThermoFisher | Cat#: 15–5921 | FACS 1:200 (RRID:AB_468811) |
| Antibody | Anti-Mouse CD140a (PDGFRA) APC (clone: APA5) | ThermoFisher | Cat#: 17–1401 | FACS 1:100 (RRID:AB_529482) |
| Antibody | Anti-Mouse CD24 APC-eFluor 780 (clone: M1/69) | ThermoFisher | Cat#: 47–0242 | FACS 1:200 (RRID:AB_10853172) |
| Antibody | Anti-Mouse CD51 PE (clone: RMV7) | ThermoFisher | Cat#: 12–0512 | FACS 1:100 (RRID:AB_465703) |
| Antibody | Anti-Mouse CD90.1 APC-eFluor 780 (clone: HIS51) | ThermoFisher | Cat#: 47–0900 | FACS 1:100 (RRID:AB_1272256) |
| Antibody | Anti-Mouse CD90.2 APC-eFluor 780 (clone: 53–2.1) | ThermoFisher | Cat#: 47–0902 | FACS 1:100 (RRID:AB_1272187) |
| Antibody | Anti-Mouse BP1 APC (clone: 6C3) | ThermoFisher | Cat#: 17–5891 | FACS 1:100 (RRID:AB_2762697) |
| Antibody | Anti-Mouse CD105 (Endoglin) Biotin (clone: MJ7/18) | ThermoFisher | Cat#: 13–1051 | FACS 1:100 (RRID:AB_466555) |
| Antibody | Anti-Mouse Tie2 (clone: Tek4) | ThermoFisher | Cat#: 14–5987 | FACS 1:20 (RRID:AB_467792) |
| Antibody | Anti-Mouse Leptin R Biotinylated Antibody (goat polyclonal) | R and D Systems | Cat#: BAF497 | FACS 1:50 (RRID:AB_2296953) |
| Antibody | Goat anti-GFP (polyclonal) | Novus Biologicals | Cat#: NB100-1770 | IF 1:200 (RRID:AB_10128178) |
| Antibody | Rabbit anti-Perilipin (polyclonal) | ThermoFisher | Cat#: PA5-72921 | IF 1:200 (RRID:AB_2718775) |
| Antibody | Rabbit anti-Osteocalcin (polyclonal) | Abcam | Cat#: ab93876 | IF 1:200 (RRID:AB_10675660) |
| Antibody | Rabbit anti-Collagen II (polyclonal) | Abcam | Cat#: ab34712 | IF 1:200 (RRID:AB_731688) |
| Antibody | Rat anti-Endomucin (Clone: V.7C7) | ThermoFisher | Car#: 14-5851-82 | IF 1:400 (RRID:AB_891527) |
| Antibody | Alexa Fluor 488 donkey anti-goat (polyclonal) | ThermoFisher | Cat#: A32814 | IF 1:500 (RRID:AB_2762838) |
| Antibody | Alexa Fluor 594 donkey anti-rabbit (polyclonal) | ThermoFisher | Cat#: A21207 | IF 1:500 (RRID:AB_141637) |

*Continued on next page*

*Continued*

| Reagent type (species) or resource | Designation | Source or reference | Identifiers | Additional information |
|---|---|---|---|---|
| Antibody | Alexa Fluor 700 goat anti-rabbit (polyclonal) | ThermoFisher | Cat#: A21038 | IF 1:500 (RRID:AB_2535709) |
| Antibody | Alexa Flour 488 goat anti-rat (polyclonal) | Abcam | Cat#: ab150157 | IF 1:500 (RRID:AB_2722511) |
| Sequence-based reagent | Oligo-dT30VN | IDT | N/A | (5'-AAGCAGTGGTA TCAACGCA GAGTACT30VN-3') |
| Sequence-based reagent | Template-switching oligonucleotide (TSO) | Exiqon | N/A | (5'-AAGCAGTGGTATC AACGCAGAGTAC ATrGrG+G-3') |
| Sequence-based reagent | ISPCR primers | IDT | N/A | (5'-AAGCAGTGGTAT CAACGCAGAGT-3') |
| Sequence-based reagent | dNTP Set (100 mM) | ThermoFisher | Cat#: 10297-018A | |
| Sequence-based reagent | ERCC (External RNA Controls Consortium) ExFold RNA Spike-In Mixes | ThermoFisher | Cat#: 4456740 | |
| Peptide, recombinant protein | SMARTScribe reverse transcriptase | Clontech | Cat#: 639538 | |
| Peptide, recombinant protein | Streptavidin PE-Cyanine7 Conjugate | ThermoFisher | Cat#: 25-4317-82 | |
| Peptide, recombinant protein | Epidermal growth factor | PeproTech | Cat#: 315–09 | |
| Peptide, recombinant protein | Platelet-derived growth factor BB | PeproTech | Cat#: 315–18 | |
| Peptide, recombinant protein | Basic fibroblast growth factor | Sigma-Aldrich | Cat#: F0291 | |
| Peptide, recombinant protein | Transforming growth factor β1 | PeproTech | Cat#: 100–21 | |
| Commercial assay or kit | KAPA HiFi Hot Start ReadyMix | Kapa Biosystems | Cat#: KK2602 | |
| Commercial assay or kit | HS NGS Fragment Kit (1–6000 bp), 1000 | Agilent | Cat#: DNF-474–1000 | |
| Commercial assay or kit | Nextera XT DNA Library Preparation kit | Illumina | Cat#: FC-131–1096 | |
| Commercial assay or kit | RNeasy Micro Kit | Qiagen | Cat#: 74004 | |
| Commercial assay or kit | RNAScope Multiplex Fluorescent V2 Assay | Acdbio | Cat#: 323100 | |
| Chemical compound | EDTA | ThermoFisher | Cat#: 15573–038 | |
| Chemical compound | Fluoromount-G | ThermoFisher | Cat#: 00-4958-02 | |
| Chemical compound | Paraformaldehyde (PFA) | Electron Microscopy Sciences | Cat#: 15710 | |

*Continued on next page*

*Continued*

| Reagent type (species) or resource | Designation | Source or reference | Identifiers | Additional information |
|---|---|---|---|---|
| Chemical compound | Matrigel | Corning | Cat#: CB40234A | |
| Chemical compound | Bovine serum albumine (BSA) | Sigma-Aldrich | Cat#: A9647 | |
| Chemical compound | Triton X-100 (10%) | ThermoFisher | Cat#: 85111 | |
| Chemical compound | Recombinant RNase inhibitor | Clontech | Cat#: 2313B | |
| Chemical compound | Saffron | Sigma-Aldrich | Cat#: S8381-5G | |
| Chemical compound | Acid Fuchsin | Sigma-Aldrich | Cat#: F8129-25G | |
| Chemical compound | Acid Red 73 | Sigma-Aldrich | Cat#: 49823–25 MG | |
| Chemical compound | Phosphotungstic Acid | Sigma-Aldrich | Cat#: P4006 | |
| Chemical compound | Hematoxylin | Sigma-Aldrich | Cat#: MHS32-1L | |
| Chemical compound | Shandon Eosin Y | ThermoFisher | Cat#: 6766009 | |
| Chemical compound | Type II collagenase | Sigma-Aldrich | Cat#: C6885 | |
| Chemical compound | 100 U/ml DNase I | Worthington | Cat#: NC9199796 | |
| Chemical compound | Fetal bovine serum (FBS) | ThermoFisher | Cat#: 16000–069 | |
| Chemical compound | Optimal Cutting Temperature compound (OCT) | ThermoFisher | Cat#: 23-730-571 | |
| Chemical compound | Penicillin-Streptomycin Solution | ThermoFisher | Cat#: 15140–122 | |
| Chemical compound | Media 199 (M199) | Sigma-Aldrich | Cat#: C6885 | |
| Chemical compound | Agencourt AMPure XP beads | Beckman Coulter | Cat#: A63882 | |
| Chemical compound | UltraPure DNase/RNase-Free Distilled Water | ThermoFisher | Cat#: 10977023 | |
| Chemical compound | TRIzol LS | ThermoFisher | Cat#: 10296028 | |
| Chemical compound | Oil Red O | Sigma-Aldrich | Cat#: O0625 | |
| Chemical compound | Alizarin Red S | Carl Roth | Cat#: A5533-25G | |
| Chemical compound | Alcian Blue 8GX | Sigma-Aldrich | Cat#: A3157 | |
| Chemical compound | Crystal Violet | Sigma-Aldrich | Cat#: C0775 | |
| Chemical compound | MCDB201 Media | Sigma-Aldrich | Cat#: M6770 | |
| Chemical compound | Dexamethasone | Sigma-Aldrich | Cat#: D-4902 | |
| Chemical compound | L-Ascorbic acid 2-phosphate | Sigma-Aldrich | Cat#: A8960 | |

*Continued on next page*

*Continued*

| Reagent type (species) or resource | Designation | Source or reference | Identifiers | Additional information |
|---|---|---|---|---|
| Chemical compound | Insulin-transferrin-selenium (ITS) mix | Sigma-Aldrich | Cat#: I3146 | |
| Chemical compound | Linoleic acid-Albumin | Sigma-Aldrich | Cat#: L9530 | |
| Chemical compound | Indomethacin | Sigma-Aldrich | Cat#: I7378 | |
| Peptide, recombinant protein | Recombinant Human Insulin | Roche | Cat#: 11376497001 | |
| Chemical compound | Isobutyl methylxanthine | Sigma-Aldrich | Cat#: I5879 | |
| Chemical compound | 3,3',5-triiodo-L-thyronine (T3) | Sigma-Aldrich | Cat#: T6397 | |
| Chemical compound | β-glycerophosphate | Sigma-Aldrich | Cat#: G9891 | |
| Chemical compound | L-thyroxine | Sigma-Aldrich | Cat#: T0397 | |
| Chemical compound | Betaine | Sigma-Aldrich | Cat#: B0300 | |
| Chemical compound | DTT (DL-dithiothreitol) 100 mM | Promega | Cat#: P1171 | |
| Software | ImageJ | NIH | http://wsr.imagej.net/distros/osx/ij152-osx-java8.zip | RRID:SCR_003070 |
| Software | FlowJo | FLOWJ LLC | https://www.flowjo.com/ | RRID:SCR_008520 |
| Software | BD FACSAria II | BD Biosciences | http://www.bdbiosciences.com/cn/home | |
| Software | Gene Expression Commons (GEXC) database | *Seita et al., 2012* | https://gexc.riken.jp | |
| Software | bcl2fastq2 2.18 | Illumina | https://support.illumina.com/sequencing/sequencing_software/bcl2fastq-conversion-software.html | RRID:SCR_015058 |
| Software | Skewer | *Jiang et al., 2014* | https://sourceforge.net/projects/skewer | RRID:SCR_001151 |
| Software | STAR 2.4 | *Dobin et al., 2013* | https://github.com/alexdobin/STAR | |
| Software | RSEM 1.2.21 | *Li and Dewey, 2011* | https://deweylab.github.io/RSEM/ | RRID:SCR_013027 |
| Software | Scanpy 1.8. | *Wolf et al., 2018* | https://github.com/theislab/scanpy | RRID:SCR_018139 |
| Software | GraphPad Prism 9.02 | GraphPad Software | http://www.graphpad.com/scientificsoftware/prism | RRID:SCR_002798 |
| Other | RNAscope Probe-Mm-Cdh13-C3 | Acdbio | Cat#: 443251 | RNAscope 1:1500 |
| Other | RNAscope Probe-Mm-Wif1-C3 | Acdbio | Cat#: 412361-C3 | RNAscope 1:1500 |

*Continued on next page*

*Continued*

| Reagent type (species) or resource | Designation | Source or reference | Identifiers | Additional information |
|---|---|---|---|---|
| Other | 4′, 6-diamidino-2-phenylindole (DAPI) | Biolegend | Cat#: 422801 | 1 ug/ml |
| Other | Propidium iodide (PI) | Biolegend | Cat#: 421301 | 1 ug/ml |

## Lead contact and material availability

Further information and reasonable requests for resources and reagents may be directed to and will be fulfilled by the lead contact Charles KF Chan (chazchan@stanford.edu).

## Animals

Mice were maintained in the Stanford University Laboratory Animal Facility in accordance with Stanford Animal Care and Use Committee and National Institutes of Health guidelines. Mice were housed in sterile micro-insulators and given water and rodent chow ad libitum. B6 mice (C57BL/Ka-Thy1.1-CD45.1; JAX:000406), purchased from the Jackson Laboratories, were used for cell isolation and gene expression/sequencing experiments. Immunodeficient NSG (NOD.Cg-Prkdcscid Il2rgtm1Wjl/SzJ; JAX:005557) mice were used as transplantation recipients for either intratibial or renal capsule transplantation of prospective stem cell populations. GFP mice (C57BL/6-Tg(CAG-EGFP)1Osb/J; JAX:003291) and Rainbow mice (Actin-CreERt Rosa[26]-Rainbow) were used for transplantation of bones and FACS-purified cell populations into NSG hosts. Male mice were used at ages stated in experiments in the 'Results' section.

## Flow cytometry and cell sorting

Flow cytometry and cell sorting were performed on a FACS Aria II cell sorter (BD Biosciences) and analyzed using FlowJo software. Long bones or renal grafts were dissected and freed from the surrounding soft tissue, which was then followed by dissociation with mechanical and enzymatic steps. Specifically, the tissue was placed in collagenase digestion buffer supplemented with DNase and incubated at 37°C for 60 min under constant agitation. After collagenase digestion and neutralization, undigested materials were gently triturated by repeated pipetting. Total dissociated cells were filtered through a 70-m nylon mesh and pelleted at 200 $\times g$ at 4°C for 5 min. Cells were resuspended in ACK (ammonium-chloride-potassium) lysing buffer to eliminate red blood cells and centrifuged again at 200 $\times$g at 4°C for 5 min. The pellet was re-suspended in 100 μl staining media (2% FBS/phosphate-buffered saline [PBS]) and stained with antibodies for at least 30 min at 4°C. The applied FACS antibodies can be found in the 'Key Resources' table. Living cells were gated for a lack of PI (propidium iodide; 1:1000 diluted stock solution: 1 μg/ml in water) fluorescence. Compensation, fluorescence-minus-one control-based gating, and FACS isolation were conducted prior to analysis or sorting using the antibody combinations as indicated in the respective figures and legends.

## Cell culture

Freshly sorted primary murine cells were used throughout this study and isolated by FACS and cultured as described before (*Chan et al., 2015*; *Ambrosi et al., 2017*). Cells were cultured in minimum essential medium alpha (MEM-α) with 10% FBS and 1% penicillin-streptomycin (ThermoFisher; Cat#: 15140–122) and maintained in an incubator at 37°C with 5% $CO_2$. For adipogenic differentiation, cells were expanded in a complex medium for 72 hr, induced for 48 hr, followed by a differentiation period of 5 days. For the expansion phase, a complex medium of 60% Dulbecco's modified Eagle medium (DMEM)-low glucose (Invitrogen) and 40% MCDB201 (Sigma) was supplemented with 100 U/ml penicillin and 1000 U/ml streptomycin (Invitrogen), 2% FBS, 1× insulin-transferrin-selenium (ITS) mix, 1× linoleic acid conjugated to BSA, 1 nM dexamethasone, and 0.1 mM L-ascorbic acid 2-phosphate (all from Sigma) and added. Before use, growth factors were added to the medium: 10 ng/ml epidermal growth factor (PeproTech), 10 ng/ml platelet-derived growth factor BB (PeproTech), and 5 ng/ml basic fibroblast growth factor (bFGF; Sigma-Aldrich). For adipogenic differentiation,

an induction medium (growth medium without growth factors) containing 5 µg/ml human insulin (Roche), 50 µM indomethacin, 1 µM dexamethasone, 0.5 µM isobutylmethylxanthine, and 1 nM 3,3′,5-triiodo-L-thyronine (T3) (all from Sigma-Aldrich) was added for 48 hr, followed by further differentiation in the growth medium without growth factors and the addition of T3 and insulin only. Oil red O staining was performed by fixing cells with 4% PFA for 15 min at room temperature. For the preparation of oil red O working solution, a 0.5% stock solution in isopropanol was diluted with distilled water at a ratio of 3:2. The working solution was filtered and applied to fixed cells for at least 1 hour at room temperature. Cells were washed four times with tap water before evaluation. To induce osteogenic differentiation, pre-confluent cells were supplemented with osteogenic medium (alphaMEM-low glucose (Invitrogen)) with 2% FBS, 100 nM dexamethasone, 0.2 mM L-ascorbic acid 2-phosphate, and 10 mM β-glycerophosphate for 14 days. Cells were then formalin-fixed and stained with 2% Alizarin red S (Roth) in distilled water. Wells were washed twice with PBS and once with distilled water. A micromass culture was used for the chondrogenesis assay. To this end, a 5 µl droplet of cell suspension (approximately $1.5 \times 10^7$ cells/ml) was pipetted into the center of a well (48-well plate). After cultivating the micromass culture for 2 hr in the incubator, warm chondrogenic media (DMEMhigh [Invitrogen]) with 10% FBS, 100 nM dexamethasone, 1 µM L-ascorbic acid-2-phosphate, and 10 ng/ml transforming growth factor β1 were added. Cell medium was changed every other day. After 21 days, cells were fixed and stained with 1% Alcian blue staining (Sigma) for 30 min at room temperature. Cells were rinsed three times with 0.1 M HCl. To neutralize acidity, a washing step with dH$_2$O was conducted before microscopic analysis. CFU-F assays were conducted by freshly sorting a defined number of cells of desired cell populations into separate culture dishes containing expansion media. The medium was changed twice a week. At day 10, cells were fixed and stained with crystal violet (Sigma).

## Single-cell clonal assay

For single-cell clonal analysis, ocSSCs or pvSSCs were freshly derived from long bones of 8-week-old B6 mice by FACS purification as explained above. Cell populations were directly sorted into 10 cm dishes at clonal density. After 10 days, clones giving rise to colonies were harvested with a cloning cylinder and re-seeded in a 96-well plate and expanded to 80–90% confluency. Each clone was then plated for tri-differentiation assays. At the end of differentiation, oil red O staining was conducted for adipogenesis, Alizarin red S staining for osteogenesis, and Alcian blue staining for chondrogenesis.

## Histology

Dissected, soft-tissue free specimens were fixed in 2% PFA at 4°C overnight. Samples were decalcified in 400 mM ethylenediaminetetraacetic acid (EDTA) in PBS (pH 7.2) at 4°C for 2 weeks with a change of EDTA twice every week. The specimens were then dehydrated in 30% sucrose at 4°C overnight. Specimens were embedded in optimal cutting temperature (OCT) compound and sectioned at 5 mm. Representative sections were stained with freshly prepared hematoxylin and eosin or Movat pentachrome.

## Immunohistochemistry

Immunofluorescence on cryopreserved, sectioned long bone and ectopic bone specimens were incubated with 3% BSA in Tris-buffered saline (TBS) for 1 hr. Then, samples were probed with primary antibody, diluted in 1% BSA/PBS, and incubated in a humidified chamber at 4°C overnight. Specimens were washed with PBS three times. Secondary antibody was applied for 15 min at room temperature in the dark. Specimens were also incubated with 1 ug/ml of DAPI for 10 min and then washed twice. The specimens were then mounted with a coverslip using Fluoromount-G and imaged with a Leica DMI6000B inverted microscope system.

## Intratibial cell injection

Sorted cell populations from GFP mice (C57BL/6-Tg(CAG-EGFP)1Osb/J) were pelleted at $5 \times 10^3$ cells and resuspended in 2 µl matrigel suspension and injected into the bone marrow cavity through the proximal articular surface of the tibia of sublethally (5 Gy) irradiated NSG mice. 4 weeks after transplantation, bones were excised, fixed, and analyzed histologically or by FACS.

## Renal capsule transplantation

SSCs were purified using FACS as described above and resuspended in 2 μl of matrigel. Cell suspension was then transplanted underneath the renal capsule of 8- to 12-week-old immunodeficient NSG mice. Injected cells developed into a graft after 4 weeks. The grafts were surgically removed for analysis via FACS and histology. For transplants of whole bones, developmentally staged embryos or pups were harvested and femurs/tibias were micro-dissected for transplantation. Bones were incubated for 4 weeks prior to surgical excision and analysis of graft.

## In vivo clonal assay

NSG mice with renal transplants of SSC populations derived from Rainbow mice were given intraperitoneal injections of 200 mg/kg of tamoxifen (Sigma-Aldrich) dissolved in corn oil 3 and 4 days after transplantation. 2-week grafts were dissected out and histologically analyzed.

## Fracture model

For the stabilized bicortical fracture model, a needle was inserted into the medullary cavity of the femur for stabilization. The fracture was induced 5 mm distal from the knee. At the indicated timepoint after fracture induction, femur bones were harvested for FACS analyses.

## Sublethal irradiation

To investigate the effects of irradiation on skeletal cell populations, mice were sublethally irradiated with one dose of 5 Gy. Mice were sacrificed 2 weeks later, and bones were dissected and processed for FACS analysis.

## Transcriptional expression profiling by microarray

Microarray analysis was performed on highly purified, double-sorted skeletal cell populations. Each cell population represents a population of cells derived from three to four separate mice pooled together. Total RNA isolation was conducted using standard column-based RNA isolation with RNeasy Micro Kit (Qiagen). RNA was twice amplified with a RiboAmp RNA amplification kit (Arcturus Engineering, Mountain View, CA, USA). Amplified complementary RNA (cRNA) was streptavidin-labeled, fragmented, and hybridized to Affymetrix 430–2.0 arrays as recommended by the manufacturer (Affymetrix, Santa Clara, CA, USA). Raw microarray data were submitted to Gene Expression Commons (GEXC) (https://gexc.riken.jp) (*Seita et al., 2012*), where data normalization was computed against the Common Reference, which is a large collection (>11,939) of mouse array experiments deposited to the National Institutes of Health Gene Expression Omnibus (NIH GEO) database. GEXC assigns a threshold value to each probeset using the StepMiner algorithm (*Sahoo et al., 2007*) and calculates a percentile value between −100% and +100% for each available gene, allowing the comparison of mouse gene expression on a normalized, continuous scale. Meta-analysis of the Common Reference also provides the dynamic range of each probe set on the array, and, in situations where there are multiple probe sets for the same gene, the probe set with the widest dynamic range was used for analysis. The Affymetrix Mouse Genome 430 2.0 Array includes 45,101 probe sets, of which 17,872 annotated genes are measurable. Microarray data are publicly accessible under https://gexc.riken.jp/models/2350.

## Smart-Seq2 single-cell RNA sequencing

Single cells were isolated via FACS as described above from freshly processed long bones pooled from four 8- to 10-week-old B6 mice. Single cells were captured in separate wells of a 96-well plate containing 4 μl lysis buffer (1 U/μl RNase inhibitor [Clontech, Cat#: 2313B]), 0.1% Triton (Thermo Fisher Scientific, Cat#: 85111), 2.5 mM dNTP (Invitrogen, Cat#: 10297–018), 2.5 μM oligo dT30VN (IDT, custom: 5'–AAGCAGTGGTATCAACGCAGAGTACT30VN-3'), and 1:600,000 External RNA Controls Consortium ExFold RNA Spike-In Mix 2 (ERCC; Invitrogen, Cat#: 4456739) in nuclease-free water (Thermo Fisher Scientific, Cat#: 10977023) according to the SmartSeq2 protocol by *Picelli et al., 2014*. Two 96-well plates per SSC population with a single cell per well were sorted and processed. Cells were spun down and plates kept at −80°C until complementary DNA (cDNA) synthesis, which was conducted using oligo-dT-primed reverse transcription with SMARTScribe reverse transcriptase (Clontech, Cat#: 639538) and a locked nucleic acid containing template-

switching oligonucleotide (TSO; Exiqon, custom: 5′-AAGCAGTGGTATCAACGCAGAGTACATrGrG +G-3′). PCR amplification was conducted using KAPA HiFi HotStart ReadyMix (Kapa Biosystems, Cat#: KK2602) with In Situ Polymerase Chain Reaction (ISPCR) primers (IDT, custom: 5′-AAGCAG TGGTATCAACGCAGAGT-3′). Amplified cDNA was then purified with Agencourt AMPure XP beads (Beckman Coulter, Cat#: A63882). After quantification, cDNA from each well was normalized to the desired concentration range (0.05–0.16 ng/μl) by dilution and consolidated into a 384-well plate. Subsequently, this new plate was used for library preparation (Nextera XT kit; Illumina, Cat#: FC-131–1096) using a semi-automated pipeline. The barcoded libraries of each well were pooled, cleaned-up, and size-selected using two rounds (0.35x and 0.75x) of Agencourt AMPure XP beads (Beckman Coulter), as recommended by the Nextera XT protocol (Illumina). A high-sensitivity fragment analyzer run was used to assess fragment distribution and concentrations. Pooled libraries were sequenced on NovaSeq6000 (Illumina) to obtain 2x101 bp paired-end reads.

## Single-cell RNA-sequencing data processing

Sequenced data were demultiplexed using bcl2fastq2 2.18 (Illumina). Raw reads were further processed using skewer for 3′ quality trimming, 3′ adaptor trimming, and removal of degenerate reads (*Jiang et al., 2014*). Trimmed reads were then mapped to the mouse genome vM20 using STAR 2.4 (*Dobin et al., 2013*), and counts for gene and transcript reads were calculated using RSEM 1.2.21 (with an average of >75% of uniquely mapped reads) (*Li and Dewey, 2011*). Data were explored and plots were generated using Scanpy 1.8 (*Wolf et al., 2018*). To select for high-quality cells only, we excluded cells with fewer than 250 genes and 2500 counts, and genes detected in less than three cells were excluded. Cells with the mitochondrial gene content higher than 10% of all expressed genes were excluded from downstream analysis as were cells with the ribosomal gene content above 5%. Additionally, if the ERCC fraction of a cell was higher than 30%, this cell was also excluded. A total of 143 ocSSCs and 169 pvSSCs passed these stringent filter criteria and were used for analysis. Raw counts per million (CPM) values were mean- and log-normalized, and then data were scaled. Principal component (PC) 'elbow' heuristics were used to determine the number of PCs for clustering analysis with UMAP (v. 0.4.6.) and leidenalg (v. 0.8.2.). Differential gene expression between ocSSCs and pvSSCs as well as Leiden clusters was calculated by Wilcoxon-Rank-Sum test. Cell cycle status was assessed using the 'score_genes_cell_cycle' function with the updated gene list provided by *Nestorowa et al., 2016*. EnrichR (*Chen et al., 2013*) was used to explore enrichment for pathways (via BioPlanet 2019) and ontologies (via GO Biological Processes 2018) of differentially expressed genes in ocSSCs and pvSSCs. All raw data and a gene table are publicly available at GEO under sample accession number GSE161477.

## RNA in situ hybridization

Fresh mouse long bone specimens (10–12 weeks) were fixed overnight before decalcification using 400 mM EDTA (Invitrogen, Cat#: 15573–038) in PBS (pH 7.2) at 4℃ for 3 weeks. Specimens were then paraffin-embedded and sectioned at 10 microns. Sections were processed for RNA in situ hybridization using RNAscope Multiplex Fluorescent Reagent Kit v2 according to manufacturer's protocol (Advanced Cell Diagnostics, Cat#: 323100). *Cdh13* (ACD, Cat#: 443251-C3) and *Wif1* (ACD, Cat#: 412361-C3) RNA probes were used. Additional immunofluorescence was added by staining with rat anti-mouse endomucin antibody (ThermoFisher, Cat#: 14-5851-82) with secondary goat anti-rat AF488 (Abcam, Cat#: ab150157) as well as counterstaining with DAPI.

## Quantification and statistical analyses

Statistical testing was performed on results with no animals or raw data points excluded. Sample sizes were not pre-determined. Data were tested for equal variances as well as normality by Shapiro-Wilk test and corrected for, if necessary, as indicated in the figure legends. Statistical analyses were performed using unpaired, two-tailed Student's t-test for comparison of experimental groups (GraphPad Prism; version 9). Statistical significance was defined as $p < 0.05$. All data points refer to biological replicates and are presented as mean + standard error of the mean (SEM) unless otherwise stated in figure legends.

## Acknowledgements

We thank C. Queen, L. Quinn, V. Ford, C. McQuarrie, T. Naik, and L. Jerabek for lab management, A. McCarthy and C. Wang for mouse colony management, and P. Lovelace, S. Weber, C. Carswell-Crumpton from the Stanford University Institute for Stem Cell Biology and Regenerative Medicine FACS core (NIH S10 RR02933801) for experimental support. Special thanks go to Stephanie Conley as well as Lolita Penland, Brian Yu, and Michelle Tan from the Chan Zuckerberg BioHub for support with single-cell RNA sequencing. We also thank M.R. Eckart and the Stanford Gene Expression Facility (PAN Facility) for contributing to this project.

## Additional information

### Funding

| Funder | Grant reference number | Author |
| --- | --- | --- |
| Deutsche Forschungsgemeinschaft | 399915929 | Thomas H Ambrosi |
| National Institute on Aging | 1K99AG066963 | Thomas H Ambrosi |
| National Institute on Aging | K99-R00AG049958-01A1 | Charles KF Chan |
| Prostate Cancer Foundation | | Charles KF Chan |
| Siebel Foundation | | Charles KF Chan |
| NIDDK | R01DK115600 | Irving Weissman |
| Heritage Medical Foundation | | Charles KF Chan |
| American Federation for Aging Research | | Charles KF Chan |
| Endowment from the DiGenova Family | | Charles KF Chan |
| National Institutes of Health | R56 DE025597 | Michael T Longaker |
| California Institute of Regenerative Medicine | CIRMTR1-01249 | Michael T Longaker |
| Oak Foundation | | Michael T Longaker |
| Hagey Laboratory | | Michael T Longaker |
| Pitch Johnson Foundation | | Michael T Longaker |
| Gunn/Olivier Research Fund | | Michael T Longaker |
| National Institutes of Health | R01 DE026730 | Michael T Longaker |
| National Institutes of Health | R01 DE021683 | Michael T Longaker |
| National Institutes of Health | R21DE024230 | Michael T Longaker |
| National Institutes of Health | R01 DE027323 | Michael T Longaker |
| National Institutes of Health | U01 HL099776 | Michael T Longaker |
| National Institutes of Health | U24 DE026914 | Michael T Longaker |
| National Institutes of Health | R21 DE019274 | Michael T Longaker |

The funders had no role in study design, data collection and interpretation, or the decision to submit the work for publication.

### Author contributions

THA, conceptualized and wrote the paper, designed all experiments, carried out most aspects of experiments and collected the data, prepared the manuscript; RS, generated single cell RNA-sequencing analysis pipeline and assisted with data analysis and interpretation; HMS, helped with data curation and formal analysis; MYH, assisted with in vitro experiments; MPM, established Methodology and assisted with in vivo experiments; LSK, assisted with in vivo experiments and data

analysis; YW, assisted with fracture experiments and microscopy; WL, helped with histology; MM, helped with conducting single cell RNA-sequencing data acquisition; NN, provided resources and methodological expertise for single cell RNA-sequencing data acquisition; ILW, helped with supervision of the project; MTL, helped with supervision of the project; CKFC, conceptualized and wrote the paper, assisted with experiment design, carried out renal capsule transplantation experiments and assisted collecting the data, assisted in preparing the manuscript and supervised the project

## Author ORCIDs
Thomas H Ambrosi  https://orcid.org/0000-0002-7149-041X
Charles KF Chan  https://orcid.org/0000-0001-6570-7574

## Ethics
Animal experimentation: Mice were maintained in the Stanford University Laboratory Animal Facility in accordance with Stanford Animal Care and Use Committee and National Institutes of Health guidelines. Mice were housed in sterile micro-insulators and given water and rodent chow ad libitum. All of the animals were handled according to approved institutional animal care and use committee (IACUC) protocols (#33042, #9999, #27683, #10266) of Stanford. The protocol was approved by the Committee on the Ethics of Animal Experiments of Stanford University (Assurance Number: A3213-01). All surgery was performed with every effort to minimize suffering.

## Decision letter and Author response
Decision letter https://doi.org/10.7554/eLife.66063.sa1
Author response https://doi.org/10.7554/eLife.66063.sa2

# Additional files

## Supplementary files
• Transparent reporting form

## Data availability
All single cell RNA-sequencing data generated in this study was deposited at GEO under sample accession number GSE161477. Microarray data is accessible under https://gexc.riken.jp/models/2350.

The following datasets were generated:

| Author(s) | Year | Dataset title | Dataset URL | Database and Identifier |
|---|---|---|---|---|
| Ambrosi TH, Chan CKF, Sinha R | 2021 | Skeletal Stem Cell Diversity in Mouse Long Bones | https://www.ncbi.nlm.nih.gov/geo/query/acc.cgi?acc=GSE161477 | NCBI Gene Expression Omnibus, GSE161477 |
| Ambrosi TH, Chan CK | 2021 | Mouse long bone ocSSC and pvSSC lineage model | https://gexc.riken.jp/models/2350 | Gene Expression Commons, gexc.riken.jp/models/2350 |

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
