## [Decision Letter]

**Acceptance summary:**

The manuscript explores the differences between two subpopulations of skeletal stem cells (SSC) that these authors previously identified: osteochondral SSC (ocSSC) and perivascular SSC (pvSSC). Intriguingly, the data presented counter the current dogma that a single parent mesenchymal skeletal stem cell (SSC) gives rise to both bone- and fat-forming cells in bone. These results have large implications in the study of human bone physiology and osteoporosis, as we do not as of yet completely understand the midlife shift in the skeleton towards bone loss and fat gain that is associated with fracture in later life.

**Decision letter after peer review:**

Thank you for submitting your article "Skeletal stem cell diversity orchestrates long bone skeletogenesis" for consideration by *eLife*. Your article has been reviewed by a Reviewing Editor, 3 reviewers and Kathryn Cheah as the Senior Editor. The following individual involved in review of your submission has agreed to reveal their identity: Vanessa Sherk (Reviewer #1).

Essential Revisions:

1. The samples sizes are very small. Although cost may have been a barrier here, please provide references to justify the sample sizes and number of replicates to lines 463-464. Were there any outcomes on which a power analysis was performed?

2. For clarity and transparency it is strongly suggested that for all bar graphs (2A, 2E, 3E, 3G, 4C, 4D, 4F, Supp 1D, S2F, S3E), please overlay them with plots of individual data points.

3. In Methods on p 23, cell culture is described. It is not clear why such a media composition was used for expansion. ITS promotes growth but insulin in it can push cells towards adipogenesis. Dexamethasone and ascorbate can push cells towards osteogenic differentiation and away from adipogenic lineage thus artificially eliminating adipogenic cells. Added growth factors likely induced expansion but they may also affect cell fate decisions. MSC/SSC are known to grow in simple DMEM or aMEM media with heat-inactivated serum, thus not depending on added growth factors or insulin.

4. For intratibial cell injections, mice were first irradiated. Irradiation is usually used when hematopoietic cell grafting is done to give preference to donor cells. Why radiation was used here is not clear. It mostly kills host hematopoietic cells but not as many host mesenchymal progenitors. It may give advantage to donor cells to expand but it also affects the niche which is critical for cell fate decisions.

5. The nomenclature of oc vs pv SSC is based on previous studies that looked at fetal, postnatal, and adult cells. Adult ocSSC are isolated from crushed bone, however it does not exclude the possibility that these ocSSC are also perivascular b/c it is unlikely that all the stroma and vessels were removed from bone before digestion. Thus it is possible that these are two subpopulations from the same compartment in the adult animal. Although it does not put the difference between these populations under question, it does raise a concern about the right nomenclature. This may also misguide future studies as they may be using perivascular location to define only one tri-lineage population and disregard oc population.

6. Documenting the perivascular origin of the pvSSC remains the key to unlocking the true significance of the findings. First and foremost is the need to show that the pvSSC derived from bone can be stained in situ in the perivascular niche inside the endochondral/marrow compartment with the same antibodies used to identify pvSSC by FACS (e.g., anti-Sca1, -Pdgfr α, CD24, etc.). In situ staining of the pvSCC for the unique FACS markers would go a long way in cinching their hypothesis for this reviewer.

7. In conjunction with the comment above, in lines 126-127 of the Results the authors should reference the use of Sca1 as a unique marker of pvSSC. This would go a long way in justifying the seemingly arbitrary decision to include, not exclude, a small fraction of Sca1-expressing pvSSC (see panel D of Figure 2). As a consequence, the statement in the Results, lines 136-137, may not be warranted. In this regard, there are a few pieces of evidence to suggest that ocSSCs and pvSSCs may be more heterogeneous than the authors suggest. Specifically, there seems to be a small population of ocSSCs that have adipogenic lineage potential, as evidenced by Figure 1e (ocSSC does still have 0.8% adipocytes) and Figure 2e (there is a small population of ocSSCs that do in fact give rise to pvSCCs/APCs in 4-week renal grafts).

8. The authors must provide some quality control data such as total reads, percent mapped, etc of the Smart-Seq2 single-cell RNA-sequencing as these quality control data are needed to confirm the reliability of the downstream bioinformatic analysis.

9. The Discussion could be strengthened by addition of comments by the authors as to whether they consider there to be plasticity between the ocSSC and pvSSC population before final commitment to a specific BSPC or APC fate.

10. The Discussion could also be strengthened by referencing a figure(s) in the statements found in lines 248-249, 251-252, 256-258 and 283-285.

11. The alternative splicing analysis was interesting, but the graphs and accompanying explanations need more detail (Figure 5G, supplemental Figure 4e.f.g.h) to help the reader understand what they are showing and why it is significant. There are other concerns in for this topic as well. Specifically, attention is needed in the Results (lines 215-226) where alternative splicing is addressed in addition to differential gene expression as a means of characterizing distinct isoforms of translatable mRNA that are specific to pvSCC and not ocSCC. Rather than relying on an unbiased mode of identifying alternatively spliced isoforms in the pvSCC and ocSSC populations, the authors have taken a candidate gene approach, showing differential gene expression of candidate genes traditionally associated with osteo- or adipo-genesis, characterizing them as "active" or "inactive", presumably based on gene tracks shared or not shared by pvSCC and ocSSC. Taking the Pth1r as an example, one does not know from the tracks if the shorter, pvSSC isoform excludes the Pth/Pthrp ligand or CAMP binding domain of the receptor required for osteogenesis; this should be reported in the text of the Results and reviewed in the Discussion.

12. The final Results section on alternative splicing may not be well developed enough to be included in this otherwise carefully constructed manuscript and its exclusion might actually strengthen the paper.

13. It looks like the authors show tracks of various mRNA variants and in some cases those variants are more abundant in either the ocSSC or the pvSSC. These data need to be interpreted with the overall expression of those genes in mind; if they can show that the total gene expression is similar then differentials in variants becomes significant. They should do this kind of analysis on all the other candidates they cite in Figure 5 and in the Supplemental Figure 4.

14. It is strongly advised that the authors refer to some key papers which show by lineage that hypertrophic chondrocytes can become not only osteoblasts, a small proportion can become adipocytes in wild-type mice (PMID: 25092332 ;PMID: 25145361; PMID: 32662900). Discussion of these papers is relevant for placing this work into context with the prior literature. The reviewers wished to point out that the ocSSC population described herein well may likely be a subset of various stem-like cells and evidence for and against this idea must be described in the discussion/conclusions. It is for this reason that clarity regarding what type of media and why that media was used (including a consideration of all the growth factors etc) be spelled out in this paper. There is a chance that the authors were selecting for osteochondral lineage using the media that they did.

*Reviewer #1:*

This paper demonstrated the presence of two types of bone progenitors with stem cell characteristics: early osteochondral and perivascular SSCs. This paper also demonstrated the differentiation potential and differing functions of these two cell populations. The strengths are the intuitive methods and approach to demonstrating the existence and function of the cell populations. The conclusions are largely supported by the data. The paper is well-written. The largest weakness relates to small sample size in some of the experiments with a weak justification for the small sample sizes (based on other papers, but no references are provided), and the lack of data transparency at times (i.e., bar graphs with SEM bars, rather than individual data or box and whisker plots). Thus, the reproducibility of the data would need to be better demonstrated.

*Reviewer #2:*

The manuscript by Ambrosi et al., explores the differences between two subpopulations of skeletal stem cells (SSC) that they previously identified, osteochondral SSC (ocSSC) and perivascular SSC (pvSSC). These populations differ in presentation of surface markers and behavior. While ocSSC give rise to osteoblasts and chondrocytes, pvSSC have tri-lineage potential and can also become adipocytes. The Authors perform elegant studies of clonogenicity of these cells in vivo as well as lineage tracing. They confirm that only pvSSC give rise to adipocytes in addition to cartilage and bone. These populations appear at different times during embryogenesis and behave differently. While ocSSC contribute to bone formation and repair, pvSSC contribute to niche formation and regeneration after irradiation. They also examined transcriptome in these populations and found differences in gene expression and in pathways while also detecting a cross-talk between these two populations. As for the weaknesses, the nomenclature of oc vs pv SSC needs to be better justified and the use of mouse irradiation for cell grafting and complex media for in vitro expansion needs to be addressed. Overall, this is a timely and important study which is elegantly done and logically written.

*Reviewer #3:*The authors are trying to prove, by way of detailed FACS analysis of mesenchymal skeletal stem cells (SSC) isolated for the prenatal, postnatal and aging mouse skeleton, that there exists not one but two primitive SSCs in mouse modeling and remodeling long bones; one of which, derived from the osteochondral niche can give rise to bone and cartilage skeletal elements and another, from a perivascular niche in and around the bone marrow, that can give rise to adipocytes in addition to bone- and cartilage-forming cells. The latter is shown to be radiosensitive, as if it arose alongside hematopoietic stem cells in the marrow, while the former is radioresistant. The presence of two functionally distinct mesenchymal progenitors in bone challenges the dogma that there is a single progenitor that can give rise to bone, cartilage and fat in bone. If the two progenitor hypothesis is true, then it is suggested that perivascular (pv) SSC and osteochondral (oc) SSC are functionally distinct. For example, the ocSSC is active during fracture healing and the pvSSC predominant in the aging mouse skeleton. The strengths of the paper are many, especially in the very detailed study with strong cell biology methods and effective use of single cell differential gene expression analysis. However, there are weaknesses in this paper. First is the need to show that the pvSSC derived from bone can be stained in situ in the perivascular niche. Second, the authors should reference the use of Sca1 as a unique marker of pvSSC. This would be very helpful for the reader to understand the logical flow of methods and reasoning used by the authors when interpreting the results. In addition, there needs to be an expanded discussion of the characterization of distinct isoforms of translatable mRNA that are specific to pvSCC and not ocSCC. If confirmed as suggested, the careful cell specific surface phenotype for the pvSCC described here will clearly advance the field. An age related, coincident decrease in bone mass and increase in fat mass is considered to be an underlying cause of skeletal fragility in humans. Knowing that this does not represent a binary event in a single mesenchymal progenitor choosing a bone or fat fate, but rather is a collaboration between a distinct pvSSC and ocSSC could change our approach from targeting a single progenitor to targeting two distinct populations of progenitors in preventing and treating age-related bone loss.

---

## [Author Response]

Essential Revisions:1. The samples sizes are very small. Although cost may have been a barrier here, please provide references to justify the sample sizes and number of replicates to lines 463-464. Were there any outcomes on which a power analysis was performed?

We thank the reviewers for this comment. As described in the Methods section we did not predetermine sample sizes by power analysis but rather relied on experience from our previous work for comparable in vivo and in vitro studies (PMID: 19078959, PMID: 25594184, PMID: 28077677). As noted correctly, we are limited in our studies to the minimal necessary number of mice due to cost and approved animal numbers. We are confident that by applying proper statistical testing on biological and experimental replicates all conclusions drawn are valid. We have updated the QUANTIFICATION AND STATISTICAL ANALYSES section to include all information on statistics for maximum transparency (lines 604-611).

2. For clarity and transparency it is strongly suggested that for all bar graphs (2A, 2E, 3E, 3G, 4C, 4D, 4F, Supp 1D, S2F, S3E), please overlay them with plots of individual data points.

We agree with the reviewers and therefore have replaced plots with bar graphs showing individual data points accordingly.

3. In Methods on p 23, cell culture is described. It is not clear why such a media composition was used for expansion. ITS promotes growth but insulin in it can push cells towards adipogenesis. Dexamethasone and ascorbate can push cells towards osteogenic differentiation and away from adipogenic lineage thus artificially eliminating adipogenic cells. Added growth factors likely induced expansion but they may also affect cell fate decisions. MSC/SSC are known to grow in simple DMEM or aMEM media with heat-inactivated serum, thus not depending on added growth factors or insulin.

We thank the reviewers for pointing out that important shortcoming in reporting the accurate culture conditions. We have indeed cultured cells in MEM-α with 10% FBS and 1% penicillin-streptomycin for initial in vitro expansion. As reported in the original manuscript, when cells were prepared for adipogenic differentiation they were switched to the more complex expansion media for the first three days before changing to adipogenic induction media. The methods have now been corrected to state the differences (lines 438-443).

4. For intratibial cell injections, mice were first irradiated. Irradiation is usually used when hematopoietic cell grafting is done to give preference to donor cells. Why radiation was used here is not clear. It mostly kills host hematopoietic cells but not as many host mesenchymal progenitors. It may give advantage to donor cells to expand but it also affects the niche which is critical for cell fate decisions.

We purposefully chose this experimental setup for several reasons. First, we conducted sub-lethal (one dose of 5 Gy) irradiation as we have observed that engraftment of SSC populations (and mesenchymal cell types in general, data not shown) is significantly enhanced under these circumstances. Our data also shows that endogenous ocSSC numbers are negatively affected by this treatment, potentially giving space to specific niches for transplanted cells to implant (Figure 3E). As a consequence, aside from higher numbers of transplanted cells present in the bone marrow cavity, this regimen also provides differentiation inducing niches as a consequence of small disruptions caused by sub-lethal irradiation. Since the purpose of the experiment was to assess differentiation capacity of the two stem cell populations in their tissue of origin, a strategy providing a microenvironment with differentiation promoting factors was chosen.

5. The nomenclature of oc vs pv SSC is based on previous studies that looked at fetal, postnatal, and adult cells. Adult ocSSC are isolated from crushed bone, however it does not exclude the possibility that these ocSSC are also perivascular b/c it is unlikely that all the stroma and vessels were removed from bone before digestion. Thus, it is possible that these are two subpopulations from the same compartment in the adult animal. Although it does not put the difference between these populations under question, it does raise a concern about the right nomenclature. This may also misguide future studies as they may be using perivascular location to define only one tri-lineage population and disregard oc population.

The described pvSSCs have been demonstrated to preferentially co-localize with blood vessels in the original work conducted in adult C57BL/6J mice (Ambrosi et al. 2017 Cell Stem Cell, PMID: 28330582). Clonal activity observed in Rainbow mice together with flow cytometric detection of ocSSCs was demonstrated for the avascular growth plate regions of 6-week-old mice (Chan et al. 2015, PMID: 25594184, Figure 1). Data from Figure 2A of this manuscript, where we microdissected skeletal regions of long bones, shows that ocSSCs are virtually absent in the blood vessel rich diaphysis region.

To further clarify in the updated version of this manuscript, we have now conducted in situ hybridization experiments using RNAscope based on specific markers for ocSSCs and pvSSCs identified in our scRNAseq dataset of this study and their enrichment in clusters 1 and 2 (Figure 5E-F). We have identified *Wif1*, a Wnt-antagonist, to be uniquely expressed by ocSSC. Expression of Wif1 might functionally serve to preserve an undifferentiated stem cell state. We also conducted co-staining with endothelial Endomucin and could not find co-localization of Endomucin-positive blood vessels and *Wif1*-expressing cells in the bone marrow. In contrast, pvSSCs express *Cdh13*, the receptor for the adipokine Adiponectin, and are abundantly found in proximity to blood vessels (Figure 5G-H). In congruence with our flow cytometric data for the SSC subsets, *Wif1* and *Cdh13* expressing cells could also be found among cells of the periosteum (Figure 5 – supplement 1F,G). This confirmed our previous reports and showed that pvSSCs are perivascular while ocSSCs reside in avascular regions.

6. Documenting the perivascular origin of the pvSSC remains the key to unlocking the true significance of the findings. First and foremost is the need to show that the pvSSC derived from bone can be stained in situ in the perivascular niche inside the endochondral/marrow compartment with the same antibodies used to identify pvSSC by FACS (e.g., anti-Sca1, -Pdgfr α, CD24, etc.). In situ staining of the pvSCC for the unique FACS markers would go a long way in cinching their hypothesis for this reviewer.

As described in the response above the in situ localization of pvSSCs has already been demonstrated in a previous paper (Ambrosi et al. 2017 Cell Stem Cell, PMID: 28330582). We now have additional RNAscope in situ data confirming the specific localization of SSC subtypes (Figure 5E-H and supplement 1F,G).

7. In conjunction with the comment above, in lines 126-127 of the Results the authors should reference the use of Sca1 as a unique marker of pvSSC. This would go a long way in justifying the seemingly arbitrary decision to include, not exclude, a small fraction of Sca1-expressing pvSSC (see panel D of Figure 2). As a consequence, the statement in the Results, lines 136-137, may not be warranted. In this regard, there are a few pieces of evidence to suggest that ocSSCs and pvSSCs may be more heterogeneous than the authors suggest. Specifically, there seems to be a small population of ocSSCs that have adipogenic lineage potential, as evidenced by Figure 1e (ocSSC does still have 0.8% adipocytes) and Figure 2e (there is a small population of ocSSCs that do in fact give rise to pvSCCs/APCs in 4-week renal grafts).

We acknowledge the concerns by the reviewers but would like to clarify. First, Sca1 is now further highlighted as a specific marker of pvSSCs (line 127). Second, considering that under homeostatic conditions Sca1 expression is quite high in pvSSCs (e.g., see Figure 1B) and Figure 2D shows a neglectable amount of Sca1 expression (<4% of a rather rare cell type) at very low levels within ocSSCs, this could rather be due to technical reasons. Similarly, as pointed out, Figure 1E and 2E show diminishing contribution of ocSSC to adipogenic cell lineages (0.8% and 0.5% respectively). Again, this rather negligible positivity could rather be due to minor contamination occurring during FACS purification of ocSSCs for the transplantation experiments.

Single cell RNAseq data underlines the unique nature of Sca1/Ly6a as a marker of pvSSCs (Figure S3D), as do the clonal differentiation assays (Figure 1F). Finally, we acknowledge that neither ocSSC nor pvSSC might be entirely homogeneous with the markers used to purify them. For example, subclustering analysis of the single cell RNAseq data by Leiden shows additional subpopulations (Figure 5). Nonetheless, using combinations of surface markers rather than single markers demonstrates a clear enrichment for higher homogeneity of SSCs. Considering technical limitations and the scope of this manuscript we will not be able to resolve this issue to its entirety. Instead, we have now addressed that in the discussion (Lines 294-297). Future studies will have to leverage the single cell data to further purify and characterize the two cell populations.

8. The authors must provide some quality control data such as total reads, percent mapped, etc of the Smart-Seq2 single-cell RNA-sequencing as these quality control data are needed to confirm the reliability of the downstream bioinformatic analysis.

We now have re-analyzed our single cell RNA-sequencing data and amended the methods section with more details on quality filtering (lines 545-592). We also have provided information of total reads, percent mapped, ERCC fraction, mitochondrial and ribosomal content, and gene counts (Figure 3 – supplement 1A-C). None of our main conclusions has changed and the stringent quality filtering criteria give us confidence the data presented faithfully reflects SSC biology.

9. The Discussion could be strengthened by addition of comments by the authors as to whether they consider there to be plasticity between the ocSSC and pvSSC population before final commitment to a specific BSPC or APC fate.

We have added thoughts on plasticity to the discussion (lines 282-285): ”Although we cannot exclude plasticity between ocSSCs and pvSSCs our experiments conducted here do not suggest interconversion between the two cell types. Yet, specific stimuli such as high levels of Bmp2 or Wnt might be able to convert pvSSCs into ocSSC like cells (Chan et al., 2015; Matsushita et al., 2020).”

10. The Discussion could also be strengthened by referencing a figure(s) in the statements found in lines 248-249, 251-252, 256-258 and 283-285.

Figure citations have been added.

11. The alternative splicing analysis was interesting, but the graphs and accompanying explanations need more detail (Figure 5G, supplemental Figure 4e.f.g.h) to help the reader understand what they are showing and why it is significant. There are other concerns in for this topic as well. Specifically, attention is needed in the Results (lines 215-226) where alternative splicing is addressed in addition to differential gene expression as a means of characterizing distinct isoforms of translatable mRNA that are specific to pvSCC and not ocSCC. Rather than relying on an unbiased mode of identifying alternatively spliced isoforms in the pvSCC and ocSSC populations, the authors have taken a candidate gene approach, showing differential gene expression of candidate genes traditionally associated with osteo- or adipo-genesis, characterizing them as "active" or "inactive", presumably based on gene tracks shared or not shared by pvSCC and ocSSC. Taking the Pth1r as an example, one does not know from the tracks if the shorter, pvSSC isoform excludes the Pth/Pthrp ligand or CAMP binding domain of the receptor required for osteogenesis; this should be reported in the text of the Results and reviewed in the Discussion.

We thank the reviewers for the shared enthusiasm for a potential involvement of alternative splicing in biological processes of SSCs. Please refer to response to comment 12.

12. The final Results section on alternative splicing may not be well developed enough to be included in this otherwise carefully constructed manuscript and its exclusion might actually strengthen the paper.

After thoughtful consideration, we agree that even though our alternative splicing results provide interesting new aspects of differences between two distinct SSC cell types it has not sufficiently evolved to manifest a significant role in the proposed processes. We agree with the reviewers and thus in order to not distract from the main conclusions of this manuscript have decided to exclude that part and rather use this data for further development in a follow-up study allowing detailed exploration of the functional role of alternative splicing in SSC biology.

13. It looks like the authors show tracks of various mRNA variants and in some cases those variants are more abundant in either the ocSSC or the pvSSC. These data need to be interpreted with the overall expression of those genes in mind; if they can show that the total gene expression is similar then differentials in variants becomes significant. They should do this kind of analysis on all the other candidates they cite in Figure 5 and in the Supplemental Figure 4.

Per reviewers’ comments the alternative splicing analysis has been removed from the current manuscript.

14. It is strongly advised that the authors refer to some key papers which show by lineage that hypertrophic chondrocytes can become not only osteoblasts, a small proportion can become adipocytes in wild-type mice (PMID: 25092332 ;PMID: 25145361; PMID: 32662900). Discussion of these papers is relevant for placing this work into context with the prior literature. The reviewers wished to point out that the ocSSC population described herein well may likely be a subset of various stem-like cells and evidence for and against this idea must be described in the discussion/conclusions. It is for this reason that clarity regarding what type of media and why that media was used (including a consideration of all the growth factors etc) be spelled out in this paper. There is a chance that the authors were selecting for osteochondral lineage using the media that they did.

We thank the reviewers for that comment. As explained in response to comment 3 we have not selected for specific cell types during in vitro expansion since no specific growth factors were added. More importantly, in vivo experiments show virtually no contribution to the adipogenic lineage by ocSSCs, specifically bones devoid of pvSSCs and with high abundance of ocSSCs presented bone marrow adipocyte free marrow (Figure 2C). We have amended the discussion to include several other papers that have described ocSSC marker expression in reporter lines enriching for SSCs as well as the literature provided here by the reviewers (lines 250-257).